# Myeloperoxidase transforms chromatin into neutrophil extracellular traps

Garth Lawrence Burn[1,6], Tobias Raisch[2,6], Sebastian Tacke[2], Moritz Winkler[1], Daniel Prumbaum[2], Stephanie Thee[3,4], Niclas Gimber[5], Stefan Raunser[2✉] & Arturo Zychlinsky[1✉]

Neutrophils, the most abundant and biotoxic immune cells, extrude nuclear DNA into the extracellular space to maintain homeostasis. Termed neutrophil extracellular traps (NETs), these protein-modified and decondensed extracellular DNA scaffolds control infection and are involved in coagulation, autoimmunity and cancer[1,2]. Here we show how myeloperoxidase (MPO), a highly expressed neutrophil protein, disassembles nucleosomes, thereby facilitating NET formation, yet also binds stably to NETs extracellularly. We describe how the oligomeric status of MPO governs both outcomes. MPO dimers interact with nucleosomal DNA using one protomer and concurrently dock into the nucleosome acidic patch with the other protomer. As a consequence, dimeric MPO displaces DNA from the core complex, culminating in nucleosome disassembly. On the other hand, MPO monomers stably interact with the nucleosome acidic patch without making concomitant DNA contacts, explaining how monomeric MPO binds to and licences NETs to confer hypohalous acid production in the extracellular space[3]. Our data demonstrate that the binding of MPO to chromatin is governed by specific molecular interactions that transform chromatin into a non-replicative, non-encoding state that offers new biological functions in a cell-free manner. We propose that MPO is, to our knowledge, the first member of a class of proteins that convert chromatin into an immune effector.

Neutrophils migrate into inflamed tissue during septic and aseptic injury[4]. Their absence leads to overwhelming infection. Their presence is biotoxic to the infectious threat and the host[5]. Neutrophils package effector proteins into membrane-enclosed granules during differentiation. The pre-synthesis of granules allows for on-demand effector functions[6]. Granule contents are delivered to various locations in the cell, fuelling neutrophil effector functions such as phagocytosis, degranulation and the formation of NETs through NETosis.

NETs are formed by modifying, decondensing and extruding chromatin into the extracellular space[7]. Once NETosis is initiated by microbial or host-derived stimuli[8,9], multiple cellular changes occur within 1–4 h, leading to the extrusion of modified chromatin into the extracellular space[10–12].

NETs are decorated with immune-related proteins that functionalize their activity, but the molecular underpinnings of how these proteins interact with NETs is unclear. MPO, a haem-family cyclooxygenase peroxidase that produces hypohalous acid, binds to NETs[13,14]. MPO products include the antimicrobial hypochlorite, which is used as household disinfectant[13,15]. MPO is very abundant, making up to 5% of total neutrophil dry cell weight[16,17]. MPO is initially produced as a pre-pro-enzyme, which is cleaved into a heavy and a light chain. These chains are covalently linked by disulfide bonds to form monomeric MPO[18,19], which subsequently dimerizes through a single additional disulfide bond. MPO monomers and dimers have identical enzymatic activity[20]. MPO synergizes independently of catalysis with serine proteases to decondense chromatin from isolated nuclei[21]. MPO directly kills bacteria trapped in NETs by producing hypohalous acid[3]. The enzymatic activity of MPO is required to initiate NETosis for certain stimuli[8].

Here we show that MPO, but not its enzymatic activity, is required for chromatin decondensation during NET formation. Importantly, individuals with MPO deficiency cannot form NETs[22]. Notably, we demonstrate the unexpected molecular mechanism by which MPO decondenses chromatin during NETosis yet also remains bound to NETs. (1) Using super-resolution microscopy, we show that MPO periodically distributes with nucleosomes along NET filaments. (2) Biochemical experiments and single-particle electron cryomicroscopy (cryo-EM) reveal that MPO binds directly to the nucleosome acidic patch through two arginine anchors, independently of its catalytic activity. (3) MPO dimerization is essential for nucleosome disassembly, a central step in chromatin decondensation during NETosis. (4) Results from in situ electron cryotomography (cryo-ET) and experiments using sputum samples from individuals with cystic fibrosis (CF) suggest that MPO–nucleosome complexes are integral to NETs in the physiological context. We demonstrate how a protein that is not associated with chromatin remodelling or transcriptional regulation binds to and modifies chromatin destined for the extracellular space.

[1]Department of Cellular Microbiology, Max Planck Institute for Infection Biology, Berlin, Germany. [2]Department of Structural Biochemistry, Max Planck Institute of Molecular Physiology, Dortmund, Germany. [3]Department of Pediatric Respiratory Medicine, Immunology and Critical Care Medicine, Charité–Universitätsmedizin Berlin, corporate member of Freie Universität Berlin and Humboldt-Universität zu Berlin, Berlin, Germany. [4]Berlin Institute of Health at Charité–Universitätsmedizin Berlin, Berlin, Germany. [5]Advanced Medical Bioimaging Core Facility, Charité–Universitätsmedizin, Berlin, Germany. [6]These authors contributed equally: Garth Lawrence Burn, Tobias Raisch. ✉e-mail: stefan.raunser@mpi-dortmund.mpg.de; zychlinsky@mpiib-berlin.mpg.de

## MPO associates with NET nucleosomes

To delineate with molecular precision how MPO is organized along NET filaments, we stimulated neutrophils from healthy human donors with the mitogen phorbol 12-myristate 13-acetate (PMA) or the microbial toxin nigericin to induce NET formation through a reactive oxygen species (ROS)-dependent and a ROS-independent pathway, respectively. Three super-resolution microscopy techniques—stimulated emission depletion microscopy (STED), stochastic optical reconstruction microscopy (STORM) and structured illumination microscopy (SIM)—all demonstrated using a verified antibody that MPO is not continuously distributed on NET filaments (Fig. 1a,b, Extended Data Fig. 1a,c and Supplementary Figs. 1 and 2). We calculated, using an autocorrelation approach, that MPO is distributed every approximately 100–300 nm along nigericin- and PMA-induced NET filaments (Fig. 1c,d, Extended Data Fig. 1b and Supplementary Fig. 3). Moreover, we induced NETs by stimulating neutrophils with Panton–Valentine leukocidin (PVL; a pore-forming toxin produced by *Staphylococcus aureus*) and show, using SIM microscopy, that MPO is also discontinuously arranged along DNA filaments (Extended Data Fig. 2).

We reasoned that the periodic binding of MPO may be underpinned by a specific molecular interaction. As the MPO staining appeared much like the iconic 'beads on a string' arrangement of nucleosomes in chromatin preparations[23], we used an antibody (PL2.3) that labels nucleosomes through a H2A–H2B–DNA conformational epitope to examine nucleosome periodicity[24]. Nucleosome periodicity faithfully recapitulated MPO periodicity (Fig. 1c,d and Extended Data Fig. 1b). To confirm the colocalization between MPO and nucleosomes, we used dual-colour SIM and STED microscopy (Fig. 1e and Extended Data Fig. 1d). MPO and nucleosomes were positively cross-correlated, indicating co-localization (Fig. 1f and Extended Data Fig. 1e). We further corroborated this finding using another antibody, 3D9, that recognizes a neo-antigen that is generated on histone H3 tails on NETs and that also cross-correlated with MPO and with the PL2.3 antibody, verifying our previous results (Extended Data Fig. 1f,g). This suggested that the positioning of MPO along NET filaments depends on nucleosomes.

To further probe MPO–nucleosome interactions, mononucleosomes from PMA-induced NETs were digested with micrococcal nuclease (which preferentially cuts internucleosomal DNA, therefore preserving nucleosomes) and fractionated using sucrose gradients. We analysed these mononucleosome fractions using DNA electrophoresis and western blotting. MPO, histones and DNA co-migrated in the same fractions, suggesting an MPO–nucleosome complex (Fig. 1g). To test this, we immunoprecipitated MPO from PMA- or nigericin-stimulated, micrococcal nuclease-digested NETs. Histones co-immunoprecipitated with MPO, revealing a direct or indirect interaction between MPO and nucleosomes (Fig. 1h). This interaction was lost when the samples were pretreated with DNase I (which processively digests DNA, leading to nucleosome disassembly), indicating that an intact nucleosome is required for the MPO–nucleosome interaction (Fig. 1h). This was recapitulated after induction with monosodium urate (MSU) crystal NET induction (Extended Data Fig. 3a).

We used native polyacrylamide gel electrophoresis shift assays to directly assess the interaction between MPO and nucleosomes. Both native MPO purified from human neutrophils (predominantly a disulfide-bonded dimer of heavy/light chain dimers; hereafter, MPO) and a monomeric recombinant MPO (rMPO; resembling a partially processed monomer of MPO of which the heavy and light chain are fused into one polypeptide chain[25]) directly interacted with HeLa mononucleosomes within the physiological range of 50–150 mM NaCl, as observed by upshifts of the DNA band (Fig. 1i and Extended Data Fig. 3b,c). Dimerization of MPO is therefore not required for the MPO–nucleosome interaction. Notably, MPO monomers and dimers exist in the blood plasma[26] and MPO monomers also exist in native MPO preparations (Extended Data Fig. 3d). The haem-containing peroxidase

catalase was used as a control and did not induce an upshift (Extended Data Fig. 3b,c). As neither the peroxidase inhibitor sodium azide nor the MPO inhibitor 4-aminobenzoic acid hydrazide (ABAH)[27] influenced this band shift, we conclude that the catalytic activity of MPO is not required for its association with nucleosomes. Conversely, MPO retains its catalytic activity when bound to nucleosomes—a further indication that nucleosome binding and enzymatic activity are independent features of the protein (Extended Data Fig. 3e).

## MPO binds to the nucleosome acidic patch

To elucidate the molecular interaction between MPO and nucleosomes, we reconstituted rMPO and a nucleosome comprising histones H2A, H2B, H3 and H4 as well as the Widom-601 DNA sequence[28] and separated the excess rMPO using size-exclusion chromatography (Supplementary Fig. 4a). Cryo-EM single-particle analysis revealed a stable complex, refined to 3.8 Å, corresponding to rMPO bound to a nucleosome (Fig. 2a and Supplementary Fig. 8).

The rMPO monomer binds only to the histone core complex, with no contacts with DNA or histone tails (Fig. 2b), in agreement with the ability of MPO to bind to nucleosomes that lack histone tails (Supplementary Fig. 4b). The main rMPO-binding site is located along and around a highly contoured cleft between H2A and H2B that is spanned by several acidic residues (Fig. 2c,d). This acidic patch[29] is a well-known binding platform for nucleosome interactors such as the yeast silencing factor Sir3 (ref. 30), the histone methyltransferase DOT1L[31] and the tail of histone H4 itself[32]. These interactors typically bind to the acidic patch using arginine side chains that are anchored at defined positions including direct contacts with six acidic residues of H2A (Glu56, Glu61, Glu64, Asp90, Glu91 and Glu92) and two of H2B (Glu105 and Lys113)[33]. In rMPO, residue Arg473 acts as an anchor, reaching deep into the canonical Arg anchor 1 site on the acidic patch and interacting with H2A residues Glu61, Asp90 and Glu92 using hydrogen bonding, and is further stabilized by hydrophobic interactions with Leu65 of H2A and Leu103 of H2B (Fig. 2d,e). Arg653 serves as the second anchor, which does not reach as deeply into the acidic patch as Arg473. Nevertheless, its interaction with H2A Glu56 as well as the hydrogen bond between Lys654 and H2B Glu110 help to stabilize the interface (Fig. 2d,f).

We compared the binding mode of rMPO with several dozen other structures of acidic-patch binders and found, in agreement with the previous literature[33], that all of them use either one or both of the arginine anchor positions observed for rMPO (several examples are shown in Extended Data Fig. 4a–f). Notably, the presence of an arginine in the anchor 1 position is widely spread, with the side-chain conformation and contacts to H2A Glu61, Leu65, Asp90 and Glu92 and H2B Leu103 being highly similar (Extended Data Fig. 4g). On the other hand, an arginine in anchor position 2 is less common and the orientations of its side chains are highly divergent and can be further classified into variant Arg type 1 and type 2, in which the guanidine head groups locate approximately to the same positions[33]. Using this nomenclature, MPO belongs to the variant Arg type 1 subclass (Extended Data Fig. 4h). The highly divergent flanking regions surrounding the arginine anchors of MPO and other acidic-patch binders suggest that they are evolutionarily unrelated and each independently acquired the ability to bind to the acidic patch.

rMPO residues Met688, Arg691 and Gln692 provide another, smaller auxiliary interface by binding to residues Gln76, Asp77 and Thr80 located on the histone H3 α1L1 elbow[33], which might contribute to the rigidity of the whole assembly (Fig. 2c,g). In essence, rMPO is a canonical acidic-patch binder, and this interaction mode is incompatible with the nucleosome stacking observed in condensed chromatin[34]. Notably, the active site of rMPO faces away from the nucleosome and does not participate in nucleosome binding (Fig. 2b), explaining why the enzymatic activity of MPO is not required for nucleosome binding[21] (Fig. 1i).

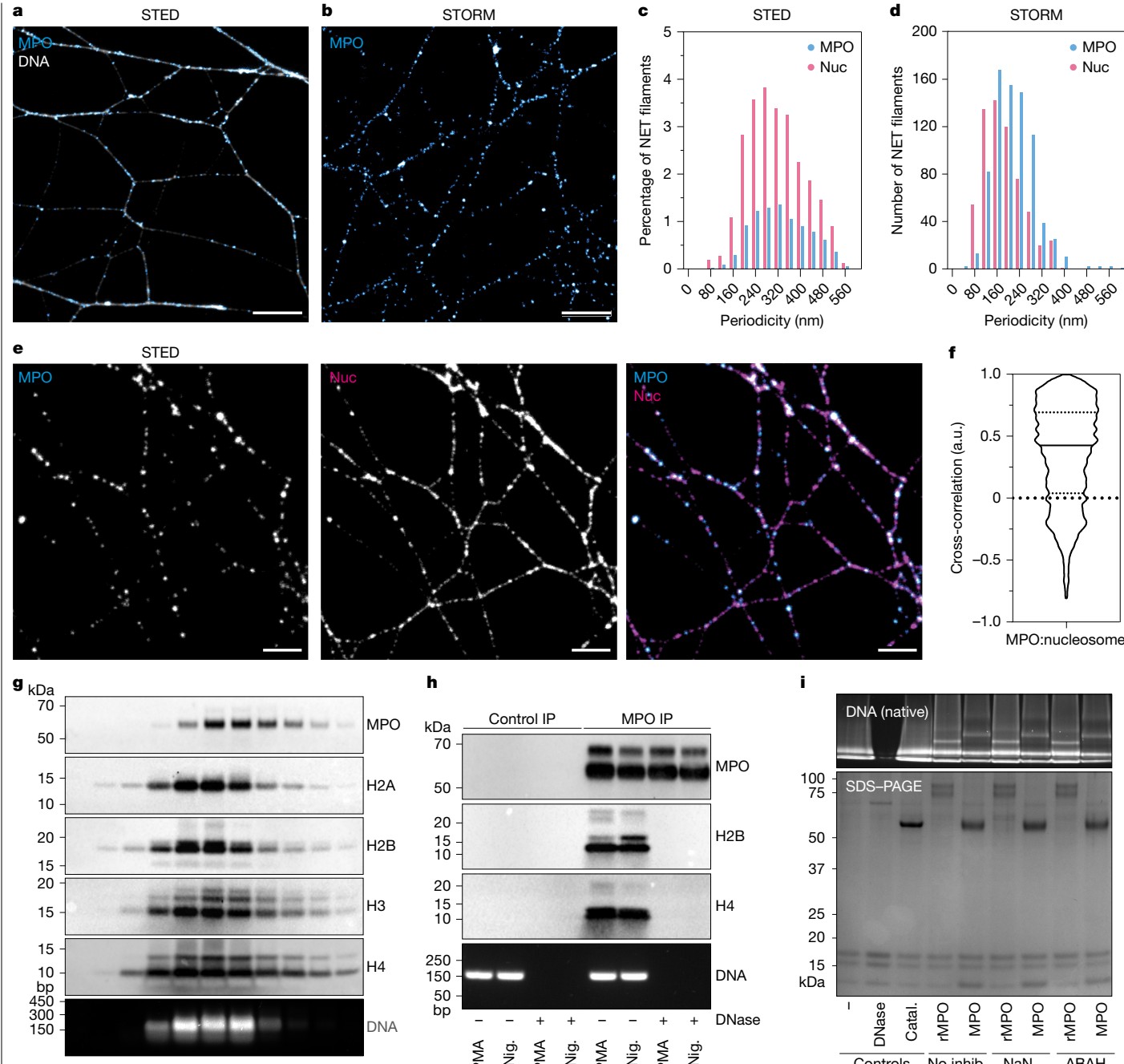

**Fig. 1 | Myeloperoxidase directly associates with nucleosomes on NETs.**
**a**, Representative STED image of NETs stained with anti-MPO and the DNA dye YOYO-1. $n = 3$ independent experiments. **b**, Representative STORM image of NETs stained with anti-MPO antibody. $n = 10$ independent experiments. **c**, Single-colour immunofluorescence and autocorrelation of MPO or nucleosomes (nuc) on the thinnest NET DNA filaments acquired using STED microscopy. $n = 3$ independent experiments (8,059 NET filaments analysed). **d**, Single-colour immunofluorescence and autocorrelation of MPO or nucleosomes on the thinnest NET DNA filaments acquired using STORM. $n = 10$ independent MPO autocorrelation experiments (1,143 NET filament fragments analysed) and $n = 5$ independent experiments (773 NET filaments analysed). **e**, Dual-colour immunofluorescence analysis of MPO and nucleosome NET staining acquired by STED microscopy. $n = 3$ independent experiments. **f**, Cross-correlation of MPO and nucleosomes quantified from **a**. $n = 3$ independent experiments (8,059 filaments analysed). a.u., arbitrary units. **g**, NET nucleosome fractionation. Neutrophils were stimulated with PMA for 4 h to induce NETs. NETs were digested into mononucleosomes using

micrococcal nuclease and fractionated over a 10–30% sucrose gradient. The fractions (fract.) were analysed using western blotting with antibodies against MPO and histones, and DNA was electrophoresed in agarose to identify 140 bp mononucleosome DNA. $n = 4$ independent experiments. **h**, Co-immunoprecipitation of MPO–nucleosome complexes from PMA- or nigericin-induced NETs. MPO immunoprecipitates were blotted for histones and MPO. DNase-I-digested nucleosomes were used as a control. Inputs were calculated from DNA concentrations of each experimental condition and monitored by agarose gel DNA electrophoresis (bottom). $n = 5$ independent experiments using PMA and nigericin (nig.) stimulation. **i**, Native gel shift assay. HeLa mononucleosomes and rMPO, MPO or catalase (catal.) were incubated 10 min at room temperature with MPO inhibitors (inhib.) azide or ABAH and subsequently subjected to native gel shift assay to monitor MPO–nucleosome interactions. SDS–PAGE gels demonstrate inputs. $n = 3$ independent experiments. Uncropped gels and blots of **g**–**i** are shown in Supplementary Fig. 22. Scale bars, 3 μm (**a** and **b**) and 1 μm (**e**).

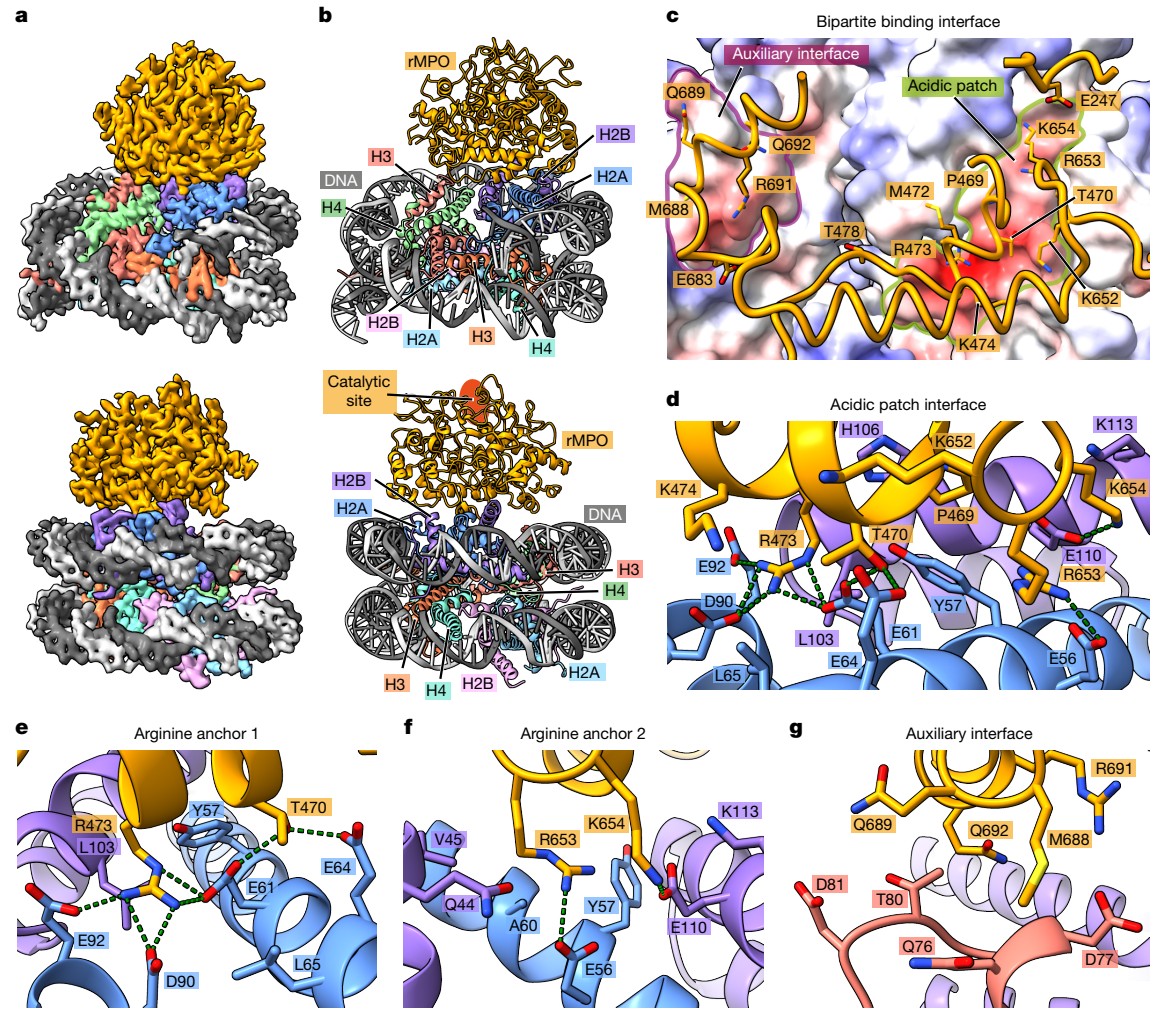

**Fig. 2 | rMPO binds to nucleosomes at the acidic patch. a**, Cryo-EM reconstruction of rMPO–nucleosome complex shown in two orientations, which are related by a turn of almost 180°. **b**, Molecular model of the rMPO–nucleosome complex in the same orientations as in **a**. The nucleosome consists of histones H2A, H2B, H3 and H4 (two copies of each), and the Widom-601 143 bp DNA sequence. **c**, A view of the surface of the nucleosome from the top through rMPO (most of the rMPO molecule is omitted to allow an unobstructed view) reveals that MPO binds predominantly to the acidic patch (demarcated in green) and also to a smaller auxiliary interface (indicated in purple). **d**, The acidic patch interface

is dominated by two arginine anchors, Arg473 and Arg653 of MPO, which bind tightly into acidic pockets generated by residues from histones H2A (light blue) and H2B (light purple). Hydrogen bonds are indicated as green dotted lines. **e**, Magnified view of arginine Arg473's canonical acidic patch anchoring interactions with Glu61, AspD90 and Glu92 of histone H2A. **f**, Magnified view of the second arginine anchor, Arg653, which interacts with H2A Glu56. **g**, The auxiliary interface is created by MPO residues Met688, Arg691 and Gln692 interacting with histone H3.

## MPO dimers dynamically bind to and destabilize nucleosomes

We tested a similar reconstitution scheme using MPO purified from human blood (Extended Data Fig. 5a). Notably, an initial cryo-EM analysis revealed mostly free DNA, probably from disassembled nucleosomes (Extended Data Fig. 5b). Similarly, in native gel shift assays using a range of MPO/rMPO–nucleosome molar ratios, only MPO but not rMPO efficiently disassembles nucleosomes (Extended Data Fig. 5c), suggesting a difference in the biochemical properties of the oligomers. Furthermore, we observed 2D classes of MPO with a proximal filamentous density that appeared to be DNA (Extended Data Fig. 5b). This finding is consistent with earlier observations that MPO can bind to DNA[35], which we confirmed by negative-stain EM and gel shift assays using only MPO and DNA (Extended Data Fig. 5d,e). To assay nucleosome disassembly, we pulled down biotinylated recombinant nucleosomes that were incubated with different concentrations of MPO or rMPO. Notably, MPO but not rMPO can disrupt and disassemble nucleosomes effectively (Fig. 3a). The addition of subequimolar amounts of MPO was

sufficient to separate histone proteins from biotinylated nucleosomal DNA, indicating unwrapping of the DNA and therefore nucleosome disassembly. By contrast, rMPO, even at high concentrations, does not efficiently disassemble nucleosomes (Fig. 3a).

To capture nucleosome disassembly and monitor possible intermediate assemblies, we cryo-plunged MPO–nucleosome samples at timepoints between 15 s and 20 min after reconstitution without size-exclusion chromatography (Extended Data Fig. 6a). All of the samples incubated for 2 min or longer contained a mixture of several molecular species, including free MPO dimers and free nucleosomes, as well as nucleosomes bound by MPO monomers and dimers, respectively (Fig. 3b,c, Extended Data Fig. 6b,c and Supplementary Figs. 10–13). The MPO monomer–nucleosome complex (the monomer is composed of the heavy and light chain that originate from the same precursor polypeptide by proteolytic cleavage and loss of residues 107–112) was refined to resolutions of between 2.9 Å and 3.0 Å (Supplementary Table 1 and Supplementary Figs. 10–13), and its structure closely resembles that of the rMPO–nucleosome complex (Supplementary Fig. 5a; root mean squared deviation (r.m.s.d.) of 0.88 Å over 1,608 residues). Indeed, MPO,

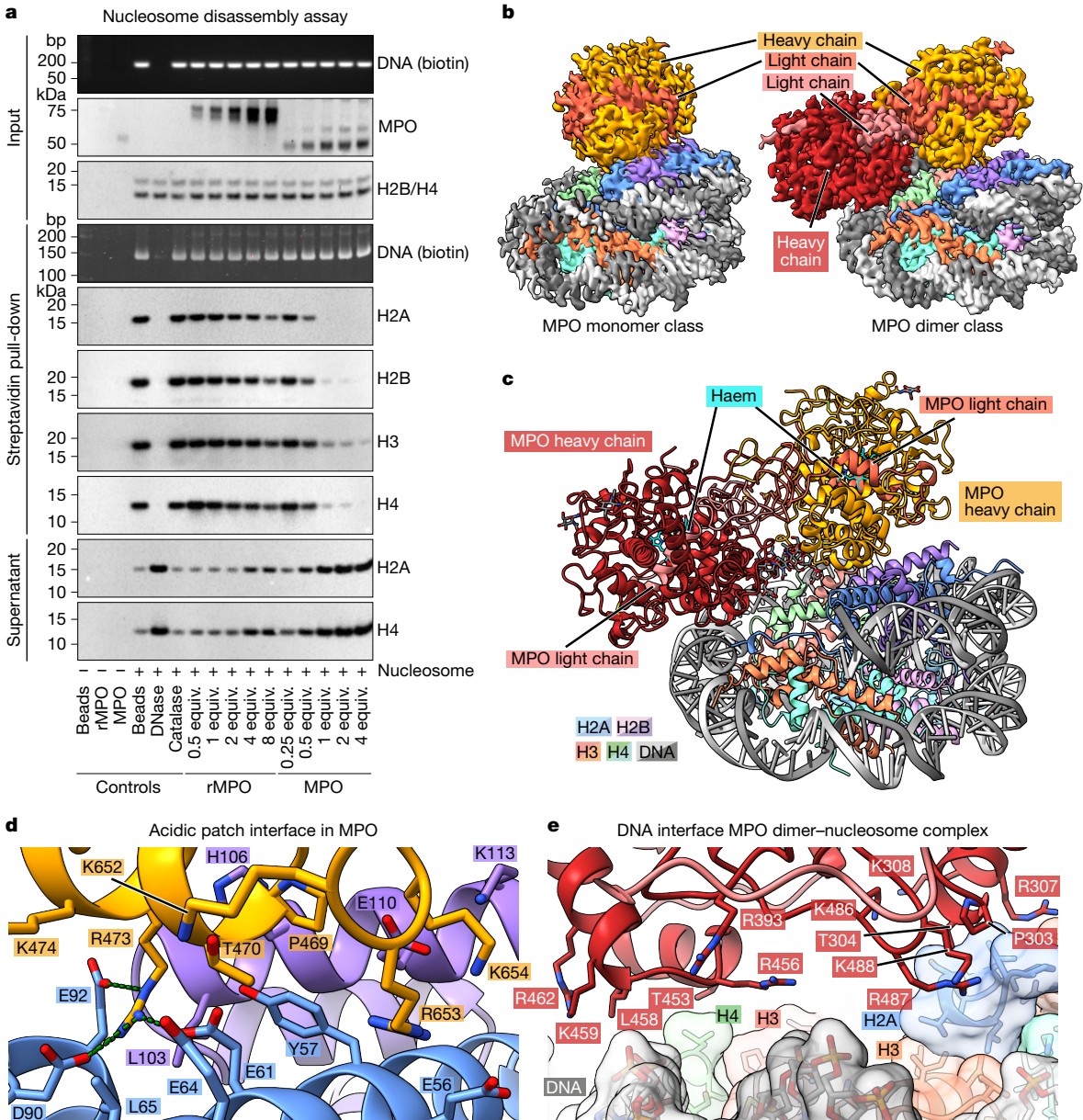

**Fig. 3 | MPO dimers bind to and destabilize nucleosomes. a**, Streptavidin pull-down of biotinylated nucleosomes incubated with indicated molar equivalents of rMPO or MPO, respectively. Nucleosomal DNA was stained on agarose gels using EtBr, while proteins in the pull-down and supernatant fractions were blotted and visualized using the indicated antibodies. Streptavidin beads alone, rMPO alone, MPO alone, DNase-I-treated nucleosomes and catalase co-incubated with nucleosomes were used as controls. $n$ = 3 independent experiments. Uncropped gels and blots are shown in Supplementary Fig. 23. **b**, Cryo-EM reconstruction of MPO monomers (left) and dimers (right) bound to nucleosomes as observed during a time-course experiment. The binding

position of MPO monomers and dimers on the acidic patch is almost identical, also to rMPO. **c**, The cryo-EM structure of MPO dimer–nucleosome complex reveals that, in addition to acidic-patch binding through the first MPO protomer, the second protomer contacts the DNA. **d**, The acidic patch interface in native MPO is highly similar to that of rMPO. Arg473 and Arg653 of MPO bind tightly into acidic pockets generated by residues from histones H2A (light blue) and H2B (light purple). **e**, Magnified view of the interface between the second MPO protomer and the DNA in the MPO dimer–nucleosome complex. The interface is characterized by charge and shape complementarity without prominent, strong contacts.

like rMPO, uses arginine residues Arg473 and Arg653 to bind to the acidic patch between H2A and H2B (Fig. 3d). Thus, rMPO and MPO both bind to nucleosomes identically, but only MPO (which is a mixture of monomers and dimers) efficiently disassembles nucleosomes (Fig. 3a).

The MPO dimer–nucleosome complex was refined to resolutions of between 3.1 Å and 3.9 Å (Supplementary Table 1 and Supplementary Figs. 9–13 and 15) and revealed that nucleosome disassembly might depend on how the MPO dimer binds to it. One of its protomers binds to the acidic patch as observed above for the monomer and rMPO (Supplementary Fig. 5a). Notably, the second protomer of the dimer

binds more peripherally and contacts the nucleosome predominantly along the DNA (Fig. 3b,c and Supplementary Fig. 5b). This interface is characterized by charge complementarity between arginines of MPO and the negatively charged DNA backbone without specific, dominant single contacts (Fig. 3e).

The MPO dimer adopts an almost identical structure and conformation to that of previously determined crystal structures of the isolated dimer (Protein Data Bank (PDB): 1MHL; Extended Data Fig. 7a). Moreover, the histones and the major portion of the nucleosomal DNA are almost identical, with free nucleosomes (Extended Data Fig. 7b)

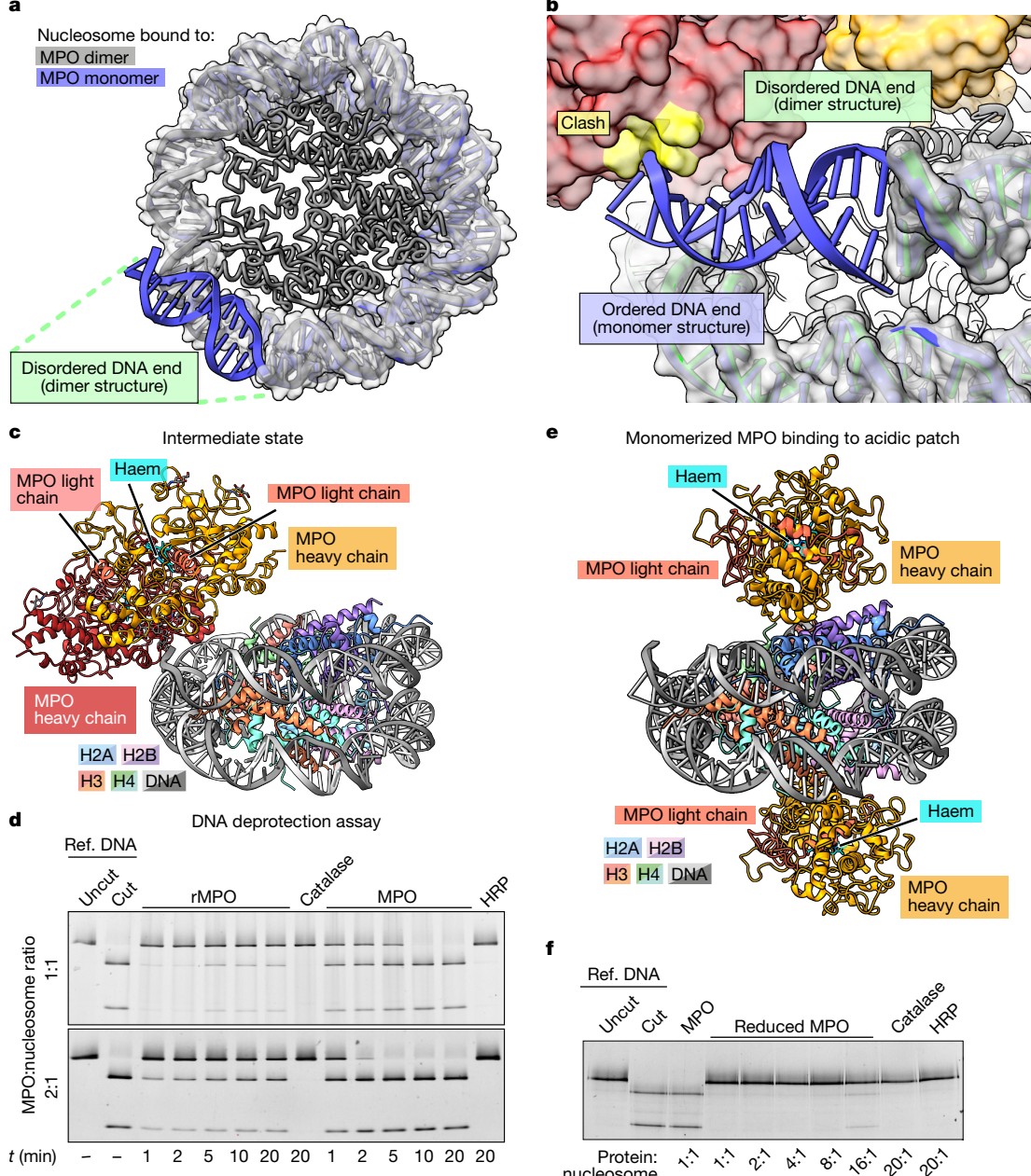

**Fig. 4 | MPO dimers displace DNA from nucleosomes. a**, Superposition of nucleosomes bound to MPO monomer (blue) or dimer (grey), respectively. Both structures are highly similar (r.m.s.d. of 0.71 Å over 1,016 residues). The major difference is the one disordered DNA end in the structure bound to the MPO dimer (highlighted in green), as visible by it protruding from the grey, transparent surface of the DNA of the nucleosome bound to MPO monomer. For clarity, the histones of the MPO monomer-bound nucleosome have been omitted. **b**, Magnified view of the superposition of MPO dimer–nucleosome and MPO monomer–nucleosome structures shows that the terminal 12 nucleotide pairs in the MPO dimer–nucleosome complex (green cartoon, and grey transparent surface) are disordered compared with the monomer–nucleosome complex (blue) to avoid clashing with the second MPO protomer (yellow surface patch). For clarity, the proteins of the monomer–nucleosome complex are omitted. **c**, Cryo-EM structure of the MPO dimer–nucleosome

complex intermediate state that was found only at the 15 s timepoint. In this state, MPO interacts only with the DNA through the arginine-rich surfaces of both protomers without contacting the histones. **d**, Nucleosome-remodelling assay using recombinant nucleosomes with an encrypted GATC restriction site that is cut by DpnII only when nucleosome–DNA interactions are perturbed. GATC nucleosomes were used to monitor the kinetics of nucleosome disassembly by rMPO and MPO. $n = 3$ independent experiments. Ref., reference. **e**, Cryo-EM structure of the nucleosome bound by two dithiothreitol (DTT)-reduced MPO monomers. The MPO molecules bind to the acidic patches on both sides of the nucleosome in an identical manner to the non-reduced sample. **f**, Nucleosome remodelling assay as described in **d**, using either non-reduced or DTT-reduced monomerized MPO. The monomerized sample cannot displace DNA from the nucleosome. $n = 3$ independent experiments. Uncropped gels of **d** and **f** are shown in Supplementary Fig. 24.

and nucleosomes bound to rMPO (Extended Data Fig. 7c) or the MPO monomer (Fig. 4a and Extended Data Fig. 7d).

Importantly, the 12 most terminal nucleotide pairs on one of the DNA ends are disordered in the structure bound by dimeric MPO compared with the other structures, in which these nucleotide pairs are ordered

(Fig. 4a,b and Extended Data Fig. 7b–d). This DNA end is in close proximity to the second MPO protomer and a superposition revealed a clash of dimeric MPO with the histone-bound conformation that the DNA adopts in complex with monomeric MPO and rMPO, respectively (Fig. 4b). Thus, the MPO dimer displaces one of the ends of the DNA,

destabilizing the nucleosome. Consistent with our biochemical findings (Fig. 3a), monomeric MPO and rMPO lack the second protomer that displaces DNA from the core complex and therefore do not destabilize the nucleosome. Importantly, the Widom-601 DNA sequence in our experiments is optimized for tight binding to the histone octamer and to prevent sliding of the DNA[28], resulting in a very compact and tight nucleosome assembly. By contrast, nucleosomes in native chromatin contain more variable DNA sequences that are less stably bound, allowing natural dynamic processes such as partial unwrapping or 'breathing' of the nucleosomal DNA[36], which can allow MPO binding to shift the equilibrium towards complete disassembly.

If the observed difference in nucleosome-disassembly activity between MPO and rMPO was caused by MPO dimerization as our data suggest, then, for longer incubation times, MPO dimer–nucleosome complexes should disappear owing to nucleosome disassembly, while MPO monomer–nucleosome complexes should remain stable. We therefore extended the incubation time of MPO and nucleosomes after mixing followed by size-exclusion chromatography (Extended Data Fig. 5a). We screened several cryo-EM grids and found that, besides disassembled nucleosomes (as reported above in Extended Data Fig. 5b), they also contained sufficient numbers of still assembled and decorated nucleosomes. We acquired a dataset of such a sample that contained only two stable molecular species, free nucleosomes and MPO monomer–nucleosome complexes (Extended Data Fig. 8a and Supplementary Fig. 14). The absence of MPO dimer–nucleosome particles in this cryo-EM dataset, while not being an ultimate proof of absence in the sample owing to the technical limitations of cryo-EM, is consistent with the hypothesis that MPO dimers are required for nucleosome disassembly. In contrast to the datasets of the 2–20 min incubations, we did not observe any MPO monomer–nucleosome complexes when we incubated them for only 15 s. Instead, along with the free nucleosome and the MPO dimer–nucleosome complex, we observed an additional, distinct conformation of dimer–nucleosome complexes (Supplementary Fig. 9). In this conformation, MPO does not contact the acidic patch, but binds to the nucleosome exclusively through the DNA, mostly by electrostatic interactions between arginine residues and the DNA backbone (Fig. 4c and Extended Data Fig. 8b). As we observed this conformation only during short incubations, we propose that it represents an intermediate state that transitions to the more stable MPO–nucleosome arrangement mediated through acidic-patch binding. The absence of MPO monomer–nucleosome complexes at this timepoint, again with the caveat that our cryo-EM analysis might in principle have missed molecular species that were under-represented in a particular sample, suggests that this complex forms more slowly. Thus, the first event when MPO encounters chromatin might be the binding to DNA, followed by recognition and binding to the nucleosome through the acidic patch, followed by DNA displacement and nucleosome disassembly.

To test the speed of nucleosome disassembly, we assayed reconstituted nucleosomes that contain a single GATC cleavage site for the restriction enzyme DpnII that is accessible only when the DNA is unwrapped from the histones. Notably, we observed DNA cleavage (and therefore nucleosome disassembly) within the first minute of incubating MPO with nucleosomes (Fig. 4d). The reaction was completed by 10 and 5 min, respectively, for 1:1 and 2:1 nMPO:nucleosome molar ratios. By contrast, in the presence of rMPO, only a minor fraction of the DNA was cleaved, consistent with our hypothesis that MPO dimers are required for DNA unwrapping. Blocking the acidic patch with the single-chain antibody fragment PL2-6 (ref. 37) (Extended Data Fig. 4a,b) prevents MPO-induced GATC cleavage by DpnII, further demonstrating that binding to the acidic patch is a prerequisite for nucleosome eviction (Extended Data Fig. 8c).

To further determine the role of disulfide-stabilized dimers for the ability of MPO to evict nucleosomes, we chemically reduced the single disulfide bridge between monomers in native MPO dimers, which efficiently dissociated the particles into stable MPO monomers that were still able to bind to nucleosomes (Extended Data Fig. 9a,b). In cryo-EM, we observed nucleosomes bound by either one or two MPO monomers or by one MPO monomer and one MPO dimer through the same acidic patch interface as in the non-reduced MPO (Fig. 4e, Extended Data Fig. 9c–e and Supplementary Figs. 17–21). This shows that MPO reduction did not change the structure of MPO monomers apart from the disulfide-stabilized light chain N terminus, which becomes disordered (Extended Data Fig. 9f), and did not change its interaction with the nucleosome. As with non-reduced MPO, only the MPO dimer but not the MPO monomer induced disorder in its neighbouring DNA terminus (Extended Data Fig. 9d), showing that this reduced MPO is a valid system to test the relevance of MPO dimers for nucleosome disassembly. We used the reduced, monomeric MPO in our GATC cleavage assay. Notably, under conditions in which non-reduced, dimeric MPO efficiently allowed DpnII cleavage of nucleosomal DNA, reduced monomeric MPO did not promote cleavage. Only at a 16-fold excess of reduced MPO did we observe faint bands corresponding to the cleavage product (Fig. 4f). This is a further indication that dimers are required for the ability of MPO to displace nucleosomal DNA and disassemble the nucleosome.

## MPO binds to nucleosomes in native NETs

To confirm the relevance of our in vitro findings in the context of NETs, we turned to cryo-ET. We stimulated neutrophils from healthy donors and acquired tomograms of central areas of NETs (Supplementary Fig. 6). Besides a highly entangled, tight network of DNA, we observed large, electron-dense granules, long protein filaments, membranous vesicles and many smaller particles in the size range of nucleosomes (Fig. 5a). We used an unbiased deep metric learning-based particle picking and clustering approach using TomoTwin[38] to identify all of the particles in the tomograms, and iteratively repeated the procedure while selecting clusters of particles with the expected size and shape of nucleosomes. After extraction and subtomogram averaging, we obtained a 31 Å reconstruction (Supplementary Fig. 7) resembling a nucleosome and featuring a significant additional density on one of its flat sides, positioned similarly to MPO in our in vitro reconstructions (Fig. 5b). Although, at the resolution of our cryo-ET reconstruction, it is challenging to definitively identify this additional density as MPO (or to distinguish between MPO monomers and dimers), the data suggest the colocalization of MPO with nucleosomes in NETs (Fig. 1e,f and Extended Data Fig. 1d–g).

Finally, we tested sputum from individuals with CF for MPO–nucleosome complexes. CF is a genetic disease that causes, among other symptoms, chronic neutrophilic inflammation of the airways[39], and the sputum is rich in NETs[40]. Sputum samples from three patients with CF contained NET-like extracellular structures containing DNA and MPO, as visualized using fluorescence microscopy (Extended Data Fig. 10a). These NET-like structures contain nucleosomes with NET-characteristic histone tail cleavages, whereas citrullinated histone H3, another marker of NETosis, is predominantly associated with chromatin still inside nuclei that were undergoing NETosis. This result suggests that the citrullinated H3 residues are cleaved off before NET extrusion into the extracellular space (Extended Data Fig. 10b,c). We digested NETs from sputum from patients with CF using micrococcal nuclease as described in Fig. 1h, and showed by MPO immunoprecipitations that histones were co-immunoprecipitated (Fig. 5c). The interaction between MPO and histones was lost after treatment with DNase I, consistent with our observation when using PMA- or nigericin-stimulated NET-derived mononucleosomes. Thus, we show that MPO interacts with nucleosomes in vivo. Notably, treatment with DNase I is a therapy in an inhalable format that effectively combats many symptoms associated with CF[40].

## Discussion

The translocation of granule-derived MPO from the cytoplasm to the nucleus can initiate NETosis[21]. Here we provide microscopy, structural

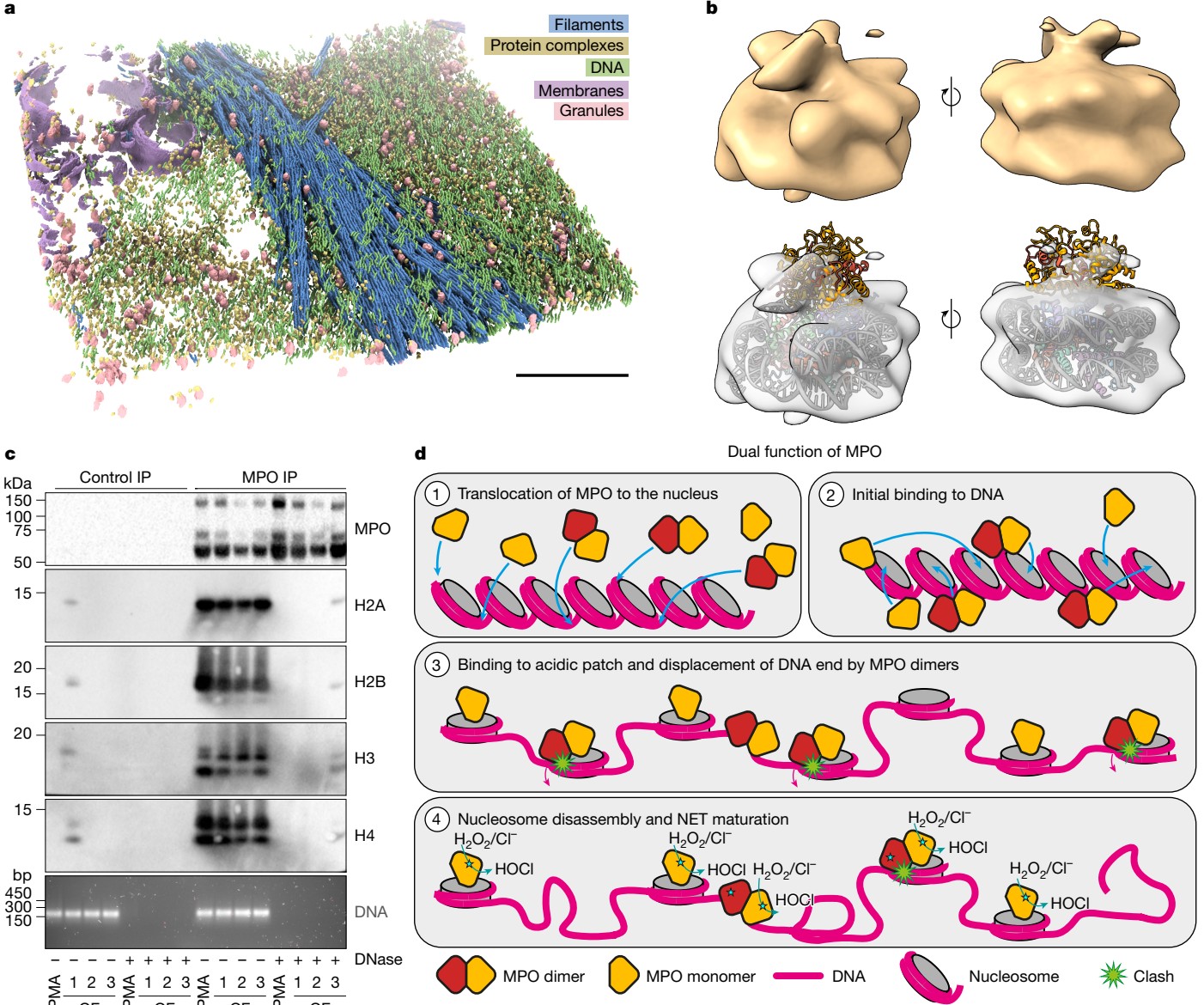

**Fig. 5 | MPO associates with nucleosomes to form NETs. a**, Representative NET segmentation of NET tomogram, showing the ultrastructural landscape of PMA-stimulated NETs. Beside broken membranes (purple), granules (pink) and unknown types of filament (blue), large protein complexes (yellow) embedded in a wide web of DNA can be seen in tomograms of PMA-stimulated NETs. **b**, Reconstruction from cryo-ET in two orientations (top). On one side of the nucleosome, an additional density is present that is remotely similar to MPO monomers as observed in the molecular model of the in vitro reconstituted MPO monomer–nucleosome complex (bottom). **c**, Co-immunoprecipitation of MPO–nucleosome complexes from PMA-induced NETs or digested mononucleosomes from the sputum of individuals with cystic fibrosis (CF). MPO immunoprecipitates were blotted for histones and MPO. DNase-I-digested nucleosomes were used as a control. Inputs were calculated from DNA concentrations of each experimental condition and were monitored by agarose

gel DNA electrophoresis (bottom). $n = 5$ independent experiments using PMA or 3 independent sputum donors with CF from different days of sample collection. Uncropped gels and blots are shown in Supplementary Fig. 25. **d**, Mechanistic model of the dual function of MPO in the context of NETs. After MPO (which is a mixture of monomers and dimers) translocates to the nucleus (1), it initially binds to chromosomal DNA (2). Then, both monomers and dimers can bind to the nucleosome acidic patch which already leads to unstacking of nucleosomes and initial chromatin decondensation (3). As dimeric MPO clashes with one end of the nucleosomal DNA, this DNA unwraps to prevent this clash, which initiates complete disassembly of MPO dimer-bound nucleosomes (4). Monomeric MPO, on the other hand, does not clash with the DNA, does not initiate nucleosome disassembly and stays attached to decondensed chromatin in mature NETs, where it produces hypochloric acid (HOCl), which is important for the activity of NETs.

and biochemical evidence that MPO performs two distinct tasks leading to the formation and effector function of NETs. We propose that MPO (1) facilitates decondensation of chromatin or (2) binds stably to nucleosomes depending on its oligomeric state and independent of hypohalous acid production.

MPO recognizes the nucleosome acidic patch using a binding mode similar to other well-characterized nucleosome interactors[33]. However, purified MPO from human neutrophils disassembled nucleosomes

without requiring ATP, which stands in stark contrast to the energy requirements of most chromatin remodellers[41]. For dimeric MPO, this specific and probably tight binding positions the second MPO protomer to interact with the DNA at the crossover site, resulting in a clash of this MPO protomer with the nearby end of the nucleosomal DNA. To compensate for this clash, the DNA end can no longer associate with the nucleosome and unwraps, ultimately destabilizing the nucleosome enough to trigger its disassembly. It is important to

point out that nucleosomes generally display some inherent structural dynamics, and DNA end unfolding can also be observed in undecorated nucleosomes without leading to complete eviction[42]. Furthermore, we cannot categorically rule out other mechanisms that might contribute to nucleosome destabilization, for example, intermediate states too transient and rare to be captured by cryo-EM. However, in all of our structures, we observed disordered ends in biochemically unstable samples and ordered ends in biochemically stable samples. The most likely explanation for these observations is that DNA end destabilization leads to nucleosome disassembly. Essentially, the MPO dimer induces molecular constraints and shifts the dynamic equilibrium of nucleosomes towards detachment of the DNA, driving nucleosome eviction (Fig. 5d).

By contrast, monomeric MPO does not clash with the DNA end and therefore does not displace DNA from the histone core complex. Instead, it even protects histones from eviction by MPO dimers as the acidic patch is preoccupied (Fig. 5d). As a consequence, MPO readily decorates NETs and may potentiate NET function through hypohalous acid production. Notably, the catalytic site of MPO is distal to the docking interface, ready to intercept substrate.

On the basis of our data, we propose that dimeric MPO decondenses chromatin during NETosis, whereas monomeric MPO interacts stably with decondensed, extruded, cell-free NETs. MPO might first bind to chromatin through weak and non-specific interactions through the DNA backbone, before locking into the acidic patch after encountering a nucleosome. We propose that, after entry of both monomeric and dimeric MPO into the nucleus when NETosis is initiated, the monomeric MPO might compete with dimeric MPO for the acidic-patch-binding site, therefore shielding some nucleosomes from disassembly and allowing for the persistence of nucleosome–MPO complexes in the extracellular space. MPO facilitates the transformation of nuclear chromatin into a non-replicative and non-encoding state that occupies the extracellular space to participate in fighting infection and promoting coagulation but also contributing to cancer and autoimmunity. Given this unexpected mechanistic role of MPO, we propose that the clinical definition of MPO deficiency, which is currently characterized by a reduction in catalytic activity, should include impairments in the ability of MPO to bind the acidic patch and evict nucleosomes.

As the MPO-induced chromatin rearrangement is probably irreversible and enables a different biological function, MPO cannot be classified as a chromatin remodeller. Instead, it is the founding member of a conceptually distinct class of chromatin factors that transform chromatin, expanding its role from a role in the storage form of genetic information into a completely unrelated role—here an important tool for immune cells. We propose that further, unidentified proteins that convert and repurpose chromatin exist in the context of immunity and also beyond, to benefit the host through the repurposing of chromatin under stress, which in turn leads to adaptation. Indeed, it is possible that MPO or other yet to be identified proteins may transform extracellular chromatin, for example, derived from necrotic cells, into a NET-like structure. Our data show a molecular mechanism by which evolutionary conserved repurposing of eukaryotic chromatin during immune responses is achieved[43,44]. Understanding what drives the generation of NETs and how proteins bind to them will instruct therapeutic interventions that block NET functions or NETosis, for example, by preventing MPO from binding to the acidic patch and blocking NET-related inflammatory processes.

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

## Methods

### Human sample collection and cell lines
Our study was conducted in accordance with the Helsinki Declaration. Anonymous blood donations from the Charité Campus Mitte blood bank and sputum samples from patients with CF were collected after obtaining informed consent. Both blood and sputum sample collection were approved by the ethics committee of Charité University Hospital, Berlin, Germany. The PLB-985 cell line (female; RRID: CVCL_2162) was donated by M. Dinauer. The PLB-985 cell line was used exclusively as a reference or control, and mycoplasma testing of the cell lines was not performed. The *MPO*-knockout PLB-985 line was generated in our laboratory[45].

### Neutrophil isolation and cell culture
Blood was collected into EDTA containing tubes, layered 1:1 on Histopaque 1119 (Sigma-Aldrich) followed by centrifugation for 20 min at 800$g$. Plasma and the upper layers of the separated blood, consisting mainly of peripheral blood mononuclear cells, were discarded. The neutrophil-rich pink layer was collected and the most dense layer consisting of red blood cells was left undisturbed. Neutrophils were washed in PBS containing 0.1% human serum albumin (HSA, Grifols), and further fractionated on a discontinuous Percoll (Pharmacia) gradient consisting of 2 ml layers with densities of 1,105 g ml$^{-1}$ (85%), 1,100 g ml$^{-1}$ (80%), 1,093 g ml$^{-1}$ (75%), 1,087 g ml$^{-1}$ (70%) and 1,081 g ml$^{-1}$ (65%). Neutrophils were carefully layered on the top of the gradient and centrifuged for 20 min at 800$g$, the interface between the 80% and 85% Percoll layers was collected and washed with PBS containing 0.05% HSA. Neutrophil purity was determined to be >95% by flow cytometry.

### NET induction
Primary neutrophils cultured in RPMI (Invitrogen) supplemented with 10 mM HEPES and 0.05% HSA were induced to form NETs using 100 nM phorbol myristate acetate (Sigma-Aldrich), 20 µM nigericin (Sigma-Aldrich), 10 nM Panton–Valentine leukocidin (IBT Bioservices) or 100 µg ml$^{-1}$ MSU crystals (Sigma-Aldrich) for at least 4 h. NET induction was checked using 1 µM Sytox Green (Thermo Fisher Scientific). After 4 h, >90% of neutrophils had undergone NETosis in all biochemical experiments performed.

### Fractionation of NET nucleosomes
The day before fractionation, continuous 10–30% sucrose gradients (1 mM EDTA, 0.5 mM EGTA, 50 mM KCl, 10% sucrose (w/v) or 30% sucrose (w/v) plus protease inhibitors) were prepared and stored at 4 °C overnight. Then, $2 \times 10^7$ neutrophils in 10 ml were plated onto 100 mm × 20 mm Petri dishes (Sarstedt) and then incubated at 37 °C under 5% CO$_2$ for 10 min before the addition of 100 nM PMA. After 4 h, the medium was gently removed, followed by one wash with 10 ml of PBS. After removing the PBS, NETs were digested on a plate with 5 ml of micrococcal nuclease buffer (10 mM Tris-HCl pH 7.5, 10 mM NaCl, 3 mM MgCl$_2$, 1 mM CaCl$_2$) containing 0.2 µl (5.58 U ml$^{-1}$) micrococcal nuclease (Thermo Fisher Scientific), protease inhibitors, neutrophil elastase inhibitor (Calbiochem) and cathepsin G inhibitor (Merck) for 8 min at room temperature. The digested reaction was stopped by the addition of 5 mM EDTA. After collecting the supernatant containing the digested nucleosomes, the preparation was centrifuged at 3,000$g$ to remove any particulate matter and then transferred to and filtered over 35 ml 100 kDa MWCO columns (Amicon). The retentate was further washed three times in micrococcal nuclease buffer with protease inhibitors and then concentrated to 0.5 ml. The nucleosome preparation was then diluted 1:1 with HEPES pH 7.5, 200 µg ml$^{-1}$ BSA and 50 mM KCl to correct for osmolality at the interface between the sample and the sucrose gradient. Nucleosomes were then layered on top of 10–30% sucrose gradients and centrifuged in a Beckman Coulter Ultracentrifuge using a SW40 rotor at 36,000 rpm for 18 h at 4 °C.

After centrifugation, the gradient was fractionated by careful pipetting from the top of the meniscus. The fractions were then split into two for western blotting or DNA agarose gel analysis.

### Protein gels, western blot and native PAGE
Protein preparations for western blotting were reduced in 1× LDS sample buffer (Invitrogen) and DTT was added at a final concentration of 100 mM before boiling at 70 °C for 15 min. The samples were run at 120 V for 1.5 h in MES running buffer (Invitrogen) using the NuPage Invitrogen Mini gel tank system in precast 4–12% gradient Bis-Tris gels. The gels were then directly stained with Instablue (Abcam) or transferred using a BioRad wet tank system onto a 0.22 µm PVDF membrane (Amersham). After transfer, the membranes were blocked for 1 h in 5% BSA followed by primary antibody overnight (Cell Signaling, mouse anti-H2A L88A6, 1:1,000; Abcam, rabbit anti-H2B, ab1790, 1:5, 000; Abcam, rabbit anti-H3, ab1791, 1:5,000; Abcam, rabbit anti-H4, ab10158, 1:5,000; MPO DAKO, A0398, 1:10,000) and then a secondary HRP conjugated antibody (Jackson Labs 1:20,000) for 1 h before washing, and the bands were developed with ECL (Pierce) using the Bio-Rad ChemiDoc. For native-PAGE, MES running buffer was replaced with native running buffer (Invitrogen) and NativePAGE gradient gels (3–12% or 4–16% Bis-Tris gels). NativePAGE sample buffer (Invitrogen) was added to protein samples before loading directly into gels and running in the cold room at 4 °C. Gels were then stained with SybrGold (Invitrogen), ethidium bromide (Sigma-Aldrich), Instablue (Abcam) or Silver stain (Thermo Fisher Scientific).

### Super-resolution microscopy
Super-resolution microscopy was performed and analysed using NanoNET as described previously[46] using identical microscopes set-ups. In brief, $1.5 \times 10^5$ freshly isolated neutrophils were seeded onto high-precision coverslips (diameter, 24 mm, 1.5H) in six-well cell culture dishes in RPMI supplemented with 0.1% HSA. NET formation was induced by incubation with 100 nM PMA or 20 µM nigericin (Sigma-Aldrich) for 4 h at 37 °C under 5% CO$_2$. The samples were then fixed in 3% paraformaldehyde (PFA) (w/v) (Electron Microscopy Sciences) for 12 min at room temperature. The coverslips were washed twice with PBS. The samples were then blocked in fish gelatin/goat serum blocking buffer for 1 h. The samples were then incubated with primary antibodies in fish gelatin/goat serum blocking buffer at 4 °C overnight (PL2.3, in-house generated (2–5 µg ml$^{-1}$); 3D9 in-house generated (10 µg ml$^{-1}$); MPO DAKO, A0398 (1:500)). After two washes with PBS, secondary antibodies (Alexa Fluor (Invitrogen) or CF dyes (Sigma-Aldrich), 1:500) and DNA dyes were added to coverslips in blocking buffer for 1 h. STORM samples were mounted on concave microscopy slides with 100 µl oxygen scavenging buffer (0.1 mg ml$^{-1}$ GLOX, 0.1 mg ml$^{-1}$ HRP, 25 mM HEPES, 5% glycerol, 25 mM glucose in PBS, pH 6.0) and sealed with dental imprint adhesive. For SIM and STED microscopy, coverslips were mounted onto microscopy slides using Prolong Gold mounting medium. Auto- and cross-correlograms were generated using NanoNET and plotted using GraphPad PRISM 5. All macros and scripts are available at GitHub (https://github.com/ngimber/NanoNET).

### Immunoprecipitation of MPO from sputum samples
Sputum samples were reduced at 37 °C by adding an equal volume of 0.1% DTT in micrococcal nuclease buffer containing protease inhibitors, neutrophil elastase inhibitor and cathepsin G inhibitor for 2 h on a rotating wheel. After 2 h, the samples were vortexed and micrococcal nuclease (30 U ml$^{-1}$) was added to digest the internucleosomal DNA for 2 h. The reaction was stopped by adding EDTA to a final concentration of 5 mM. After centrifuging the samples at 2,000$g$ for 20 min, the soluble fraction was collected, washed three times and concentrated over 100 kDa MWCO columns followed by fractionation over sucrose cushions. Only mononucleosomes were collected after fractionation on

sucrose cushions containing no EDTA or EGTA. DNase I buffer was added to pooled, collected mononucleosome fractions and the sample was then split into two and DNase I (Thermo Fisher Scientific) was added to one of the samples as a control. The samples were then incubated with 2 µg anti-MPO (DAKO) overnight at 4 °C followed by the addition of 20 µl magnetic protein G beads (Invitrogen) for 30 min at room temperature. Magnetic separation of the beads was performed followed by four washes (150 mM NaCl, 1% Triton X-100, 50 mM Tris-HCl pH 7.2, 0.15% BSA, 15% (v/v) glycerol, protease inhibitors), and the beads were then eluted in LDS sample buffer for 15 min at 70 °C. Inputs were calculated as the total amount of DNA per sample after fractionation and before DNase I digests using the Qubit double-stranded DNA quantification assay (Invitrogen). The samples were then western blotted and probed with antibodies against histones or MPO. DNA inputs were monitored and visualized by agarose gel electrophoresis and ethidium bromide staining.

## MPO–nucleosome electrophoretic shift assays

HeLA mononucleosomes or recombinant biotinylated nucleosomes (Epicypher, 16-0002 (HeLa mononucleosomes), 16-0006 (recombinant wild type) and 16-0027 (recombinant tailless)) were co-incubated with either recombinant MPO (RnD systems) or native MPO (Sigma-Aldrich) at a molar ratio of 1 nucleosome to 0.5 rMPOs or 0.25 nMPOs for 5 min at room temperature in chromatin remodelling buffer (12 mM HEPES, 40 mM Tris-HCl, 0.32 mM EDTA, 3 mM $MgCl_2$, 10% glycerol, 0.02% Igepal, 60 mM KCl and indicated NaCl concentrations, pH 7.4) and then directly loaded into native gels to monitor binding of MPO to nucleosome after the addition of native sample buffer (Invitrogen). Tris-HCl, NaCl, KCl and $MgCl_2$ were not added to the buffer for salt-less conditions. Sodium azide (Roth) and ABAH (Merck) were used as inhibitors of MPO catalytic activity and all stocks were checked for inhibitor activity before aliquoting. DNase I digestion of nucleosomes acted as a fiducial in native gels and catalase was used as a control for a non-nucleosome binding NET protein. For experiments at higher MPO:NUC ratios, incubation times were increased to 10 min. All of the experiments were performed in a 20 µl reaction in which the concentration of nucleosomes was fixed at 500 nM and the MPO molar ratios were calculated accordingly.

## Nucleosome pull-down

Biotinylated nucleosomes (Epicypher, 16-0006) were resuspended in 100 µl chromatin remodelling buffer at 1 ng µl⁻¹. rMPO and native MPO were then added to the nucleosomes at different molar ratios. A small aliquot from each reaction was used to monitor inputs. After 20 min, 400 µl of chromatin remodelling buffer containing 40 µl of magnetic streptavidin hydrophilic beads (New England Bioscience) were then added to each sample and the samples were incubated for a further 10 min on a rotating wheel. The samples were then magnetically separated and the post-pull-down supernatants were retained for analysis. The samples were then subjected to four 0.5 ml washes in chromatin remodelling buffer containing 350 mM salt. The post-wash supernatants were also retained and pooled with the post-pull-down lysates. The pooled samples were then desalted in Zeba Spin Columns (Thermo Fisher Scientific) and protein was precipitated on ice for 1 h by adding trichloroacetic acid to a final concentration of 20% (w/v). The precipitated protein was washed three times in ethanol and once in acetone and dissolved in LDS-sample buffer before being boiled at 70 °C for 10 min. The beads were then eluted in LDS sample buffer at 70 °C for 10 min. The samples were then analysed by western blotting and agarose gel DNA electrophoresis.

## GATC nucleosome remodelling assay

Chromatin remodelling buffer without EDTA from nucleosome shift assays was used as a reaction buffer. GATC nucleosomes were prepared as a 4× 400 nM stock solution and 5 µl of this preparation was pipetted into a 1.5 ml low-bind reaction tube (Eppendorf). For experiments using the PL2-6 scFv antibody fragment (Creative Biolabs), the scFv fragment was added directly to the nucleosome stock solution at a molar ratio of 3:1 leading to a final 4× stock solution of 1,200 nM PL2-6:400 nM nucleosome, which was incubated at room temperature for 30 min before performing the rest of the experimental protocol. DpnII (NEB) was prepared as a 4× 10 U µl⁻¹ stock solution and 5 µl was added to the GATC nucleosomes. A 2× stock of MPO at molar ratios corresponding to 1:1 or 1:2 nucleosome to MPO (for rMPO this corresponded to 200 nM and 400 nM solutions and for native MPO this corresponded to 100 nM and 200 nM to correct for absolute protein). To start the reaction 10 µl of MPO was added to GATC nucleosomes and to quench the reaction at various timepoints 20 µl of 2× quench buffer (10 mM Tris pH 7.4, 40 mM EDTA, 0.6% SDS and 50 µg ml⁻¹ proteinase K) and the samples were then immediately incubated at 55 °C for 30 min to remove proteins before running DNA on native polyacrylamide gels and visualizing with SybrGOLD. Catalase or horseradish peroxidase was used as a control. The GATC restriction site within the Widom-601 sequence is highlighted in bold in the following sequence: GAACCAATGGGACCATGCTTCACACCGATATCATCGCTTATGTGTTGA ATTCATCAGAATCCCGGTGCCGAGGCC**GATC**AATTGGTCGTAGACAG CTCTAGCACCGCTTAAACGCACGTACGCGCTGTCCCCCGCGTTTTAACC GCCAAGGGGATTACTCCCTAGTCTCCAGGCACGTGTCAGATATATACA TCGATGATGATGGATAGATGGATGATGGATGGATGGATGATGATGGATG AATAGATGGATGGATGAAGCTTT.

## Sample preparation and cryo-EM data acquisition of recombinant MPO in complex with nucleosomes

Nucleosomes comprising *Xenopus laevis* H2A, H2B, H3 and H4 histones and Widom-601 145 bp DNA at 4.3 mg ml⁻¹ (~22 µM) were a gift from A. Musacchio, M. Pesenti and D. Vogt; reconstitution was carried out as described previously[47]. Recombinant MPO was obtained from bio-techne/R&D Systems (cat. no. 3174-MP) and was dissolved at 2.5 mg ml⁻¹ (~31 µM) in PBS. Both were mixed at final concentrations of 1 mg ml⁻¹ nucleosomes (~5 µM) and 2 mg ml⁻¹ rMPO (~25 µM), incubated on ice for 30 min and applied to a Superdex 200 5/150 Increase column (Cytiva), which was connected to an Äkta Micro FPLC system (Cytiva) and equilibrated with 10 mM HEPES pH 7.5, 100 mM NaCl. The peak fractions were pooled and concentrated to 40 µl.

Next, 4 µl of the sample was applied to glow-discharged UltrAuFoil R 1.2/1.3 300 grids (Quantifoil) and plunge-frozen in liquid ethane using a Vitrobot Mark IV (Thermo Fisher Scientific). Cryo-EM data were acquired on a 200 kV Talos Arctica microscope (Thermo Fisher Scientific) equipped with a field emission gun at a nominal magnification of 120,000x. A total of 4,573 micrograph movies was recorded on a Falcon III camera (Thermo Fisher Scientific) operated in linear mode at a pixel size of 1.21 Å px⁻¹. A total exposure of 56 e⁻ Å⁻² was distributed over 40 frames. Details of data acquisition can be found in Supplementary Table 1.

## Sample preparation and cryo-EM data acquisition of native MPO in complex with nucleosomes

Native MPO from human leukocytes was purchased from Sigma-Aldrich (cat. no. 475911) and dissolved at 2.5 mg ml⁻¹ (~17 µM) in 10 mM HEPES pH 7.5, 100 mM NaCl. This sample was mixed with H3 601 nucleosomes (see above) at final concentrations of 1.9 mg ml⁻¹ MPO (~13 µM) and 1.1 mg ml⁻¹ nucleosomes (~5.6 µM) and kept on ice.

For the time-course experiment, 4 µl of the sample was directly applied to glow-discharged UltrAuFoil R 1.2/1.3 300 grids (Quantifoil) after 15 s, 2 min, 5 min, 10 min and 20 min, respectively, and plunge-frozen into liquid ethane using a Vitrobot Mark IV (Thermo Fisher Scientific). Data were acquired on a Cs-corrected 300 kV Titan Krios G2 (Thermo Fisher Scientific) equipped with a field emission gun. For each timepoint, ~5,000–6,000 micrograph movies were recorded in super-resolution mode (super-resolution pixel size 0.34 Å px⁻¹) on a

K3 camera (Gatan) at a nominal magnification of 105,000×. 53–54 e⁻ Å⁻² was distributed over 60 frames, and the slit width of the Bioquantum electron filter (Gatan) was set to 15 eV.

For the long incubation, the sample was kept on ice for 30 min and then applied to a Superdex 200 5/150 Increase column (Cytiva) which was connected to an Äkta Micro FPLC system (Cytiva) and equilibrated with 10 mM HEPES pH 7.5, 100 mM NaCl, analogously to the sample with rMPO (see above). The peak fractions were pooled and concentrated to 40 μl. Then, 4 μl of the sample was applied to glow-discharged UltrAuFoil R 1.2/1.3 300 grids (Quantifoil) and plunge-frozen in liquid ethane using a Vitrobot Mark IV (Thermo Fisher Scientific). Cryo-EM data were acquired on a 200 kV Talos Arctica microscope (Thermo Fisher Scientific) equipped with a field emission gun at a nominal magnification of 120,000×. A total of 2,723 micrograph movies was recorded on a Falcon III camera (Thermo Fisher Scientific) operated in linear mode at a pixel size of 1.21 Å px⁻¹. A total exposure of 56 e⁻ Å⁻² was distributed over 40 frames.

Details of data acquisitions are provided in Supplementary Figs. 9–14 and Supplementary Table 1.

### Cryo-EM data processing

Micrograph movies of the sample comprising rMPO and nucleosomes were pre-processed using cryoSPARC live[48], including patch motion correction, patch CTF estimation, particle picking using a Gaussian blob and particle extraction. An initial 2D classification was then performed using a subset of 200,000 particles, followed by ab initio reconstruction of three models using 132,381 particles associated to good 2D classes. The three ab initio models and all 5,847,459 extracted particles were then applied to heterogeneous refinement in cryoSPARC[48]. An initial homogeneous refinement using the 2,029,928 particles assigned to the best-defined class yielded a resolution of 4.0 Å. The particles were then polished and CTF parameters refined in RELION (v.3.1)[49]. Two more rounds of ab initio modelling and heterogeneous refinement with 3 and 5 classes, respectively, were performed in cryoSPARC using the shiny particles, yielding a final subset of 663,555 particles. Using these, a non-uniform refinement resulted in a reconstruction at 3.76 Å that was sharpened in PHENIX[50] by applying a sharpening B-factor of 252.6 Å². Details about data processing are provided in Supplementary Table 1 and Supplementary Fig. 8.

Processing of the 5 min timepoint of the sample including native MPO and nucleosomes was performed completely in cryoSPARC. The 5,540 micrograph movies were subjected to patch motion correction (involving twofold binning from super-resolution to native pixel size) and patch CTF estimation. In total, 1,380,469 particles were picked using a Gaussian blob picker with a diameter range of 140–200 Å. Particle picks were inspected using the Inspect Particles tool and 975,892 particles with an NCC score of above 0.190 as well as a local power score of between −2,881 and +1,642 were retained. Of these, 786,181 particles were extracted with a box size of 200 × 200 px after twofold binning and subjected to 2D classification with 150 classes. The 59 well-defined classes were used as templates for optimized particle picking, resulting in 3,659,119 particles after picking, 2,441,110 particles after inspection and 2,098,075 twofold binned particles after extraction. These extracted particles were again 2D classified. Ab initio reconstructions with five models were calculated using the 1,328,276 particles assigned to good 2D classes as well as ab initio reconstructions with three models using 769,799 particles assigned to bad 2D classes.

Subsequently, heterogeneous refinement using all extracted particles and a total of 6 ab initio models (four from the particle subset from good 2D classes and two from bad 2D classes) as reference volumes. This heterogeneous refinement yielded two junk classes corresponding to 3.0% and 19.9% of particles, a class of MPO dimers corresponding to 25.7% of all particles (which suffered from severe preferred orientation that precluded any further high-resolution reconstructions), 22.1% of particles corresponding to a free nucleosome class and classes of nucleosomes bound to MPO monomers (23.2% of particles) and dimers

(14.2% of particles), respectively. The particle subsets corresponding to free nucleosomes, monomer–nucleosome and dimer–nucleosome, respectively, were separately subjected to non-uniform refinement, followed by reference-based motion correction (all three subsets were combined in the same run). After this, they were separately refined using non-uniform refinement, local CTF refinement and another round of non-uniform refinement. This yielded final global reconstructions at 2.79 Å (nucleosome), 2.89 Å (MPO monomer–nucleosome) and 3.12 Å (MPO dimer–nucleosome), respectively. Details of data processing are provided in Supplementary Table 1 and Supplementary Fig. 11.

In the case of the dimer–nucleosome complex, refinement was finalized by a scheme of particle subtraction and local refinement with either deleting MPO (refinement centred on nucleosome resulting in 3.01 Å reconstruction) or the nucleosome (refinement centred on MPO resulting in 2.98 Å reconstruction) (Supplementary Fig. 15a). Both focused maps were combined using the volume maximum command in UCSF ChimeraX[51].

Processing of the datasets corresponding to the 2 min, 10 min and 20 min timepoints was similar to the 5 min dataset. The exact details are provided in Supplementary Figs. 10, 12 and 13 and Supplementary Table 1.

For the 15 s dataset, the initial processing strategy was also similar to the 5 min dataset up to the first heterogeneous refinement (Supplementary Fig. 9). Subsequently, three out of the six classes corresponding to free nucleosome (25.7% of all particles), MPO dimer–nucleosome (17.2% of particles) and a second MPO dimer–nucleosome arrangement (intermediate state of MPO dimer–nucleosome; 15.2% of all particles) were separately subjected to non-uniform refinement, followed by reference-based motion correction (all three subsets were combined in the same run). Next, they were separately refined using non-uniform refinement, local CTF refinement and another round of non-uniform refinement. This yielded final global reconstructions at 3.11 Å (nucleosome), 3.51 Å (MPO dimer–nucleosome) and 3.58 Å (MPO dimer–nucleosome, intermediate state), respectively. Details of data processing are provided in Supplementary Table 1 and Supplementary Fig. 9.

For the MPO dimer–nucleosome intermediate state, refinement was finalized by a scheme of particle subtraction and local refinement with either deleting MPO (refinement centred on nucleosome resulting in 3.52 Å reconstruction) or the nucleosome (refinement centred on MPO resulting in 3.87 Å reconstruction) (Supplementary Fig. 15b). Despite the lower nominal resolution, the reconstruction focused on MPO dimer was much better defined compared with the global refinement. Both focused maps were combined using the 'volume maximum' command in UCSF ChimeraX[51].

The dataset of native MPO and nucleosomes after 30 min incubation and size-exclusion chromatography was completely refined in cryoSPARC, by initial patch motion correction, patch CTF estimation and extraction of 1,364,624 particles (unbinned, 256 × 256 pixels) that had been picked using the blob picker with a radius of 140–200 Å. 2D classification with 150 classes yielded 45 well-defined classes that were used as templates for optimized picking using the template picker, resulting in 3,728,694 picked and 3,284,273 extracted particles (unbinned, 256 × 256 pixels). In parallel, ab initio reconstructions with four models were calculated using the 596,787 particles assigned to good 2D classes as well as ab initio reconstructions with four models using 767,837 particles assigned to bad 2D classes. Heterogeneous refinement was performed using the template-picked particles and three initial models from the good 2D classes as well as three models from the bad 2D classes. Then, the 765,797 particles associated to the three good classes of the heterogeneous refinement were subjected to another round of ab initio modelling (5 classes) and heterogeneous refinement (4 of the ab initio models). Two out of the four classes, corresponding to free nucleosomes (837,397 particles) and MPO monomer–nucleosome (863,330 particles), were refined by non-uniform refinement to 4.04 Å and 3.94 Å, respectively. See also Supplementary Fig. 14.

The local resolution of all final, deposited reconstructions is shown in Supplementary Fig. 16.

### Model building

For the rMPO–nucleosome, a model of rMPO (PDB: 6AZP, chain A)[52] as well as a nucleosome model comprising the *X. laevis* H2A, H2B, H3 and H4 histones and Widom-601 147-bp DNA (PDB 6R1T)[53] were rigidly docked into the reconstructed map and manually adjusted in COOT[54]. The model was optimized by iterative cycles of model adjustment in COOT and real-space refinement in PHENIX[50].

For the free-nucleosome model, PDB 6R1T was used as an initial model as well and optimized by manual adjustments in COOT and using real-space refinement in PHENIX. For the MPO dimer–nucleosome complex (main arrangement), this adjusted nucleosome was used as the initial model along with the crystal structure of the MPO dimer (PDB 1MHL)[55]; again, model was optimized using COOT and PHENIX. Deletion of one MPO monomer (containing heavy and light chain) from this MPO dimer–nucleosome model yielded the initial model for the MPO monomer–nucleosome complex, which was further optimized by COOT and PHENIX. Finally, the MPO dimer–nucleosome intermediate state model was also initiated by placing the refined nucleosome model and PDB 1MHL into the map resulting from combining both focused maps (see above) and finalized in COOT and PHENIX.

Model statistics were calculated using the MOLPROBITY[56] implementation in PHENIX and can be found in Supplementary Table 1.

### MPO–DNA-binding assay

Nucleosomal DNA from HeLa or recombinant (Widom-601 sequence) mononucleosomes were extracted using the QIAquick PCR purification kit (Qiagen). A serial dilution series of HeLa or Widom-601 nucleosomal DNA (1,600–25 nM) in chromatin remodelling buffer was then performed and rMPO or MPO was added to nucleosomal DNA at a final concentration of 200 nM in a total volume of 10 μl followed by a 30 min incubation at room temperature. Then, 10 μl of 2× native-PAGE sample buffer was added to each sample and 5 μl of each sample was then subjected to agarose DNA electrophoresis. DNA was visualized by ethidium bromide staining.

### Negative-stain EM

Purified native MPO (11 μM) was mixed with Widom-601 DNA (3 μM) and incubated for 15 min on ice. The sample was then diluted 60-fold and applied to glow-discharged copper grids coated with 8 nm continuous amorphous carbon. After 1 min incubation, excess sample was removed by blotting, the grid washed three times with Tris-buffered saline and once with 0.75% (w/v) uranyl formate solution before incubation for 1 min with 0.75% (w/v) uranyl formate. After blotting and drying the grid, the sample was analysed in a Tecnai Spirit electron microscope (Thermo Fisher Scientific) operated at 120 kV. Data were recorded at 59,000× magnification on a TemCam F416 camera (TVIPS), resulting in a pixel size of 1.67 Å

### Chemical reduction of native MPO and mass photometry of reduced and non-reduced samples

For mass photometry experiments, 100 nM native MPO was incubated at room temperature with 10 mM or 50 mM DTT in a buffer composed of 20 mM HEPES pH 7.5 and 200 mM NaCl. Mass photometry measurements were performed on a Refeyn TwoMP instrument. For this, the samples were diluted 1:10 in a drop of freshly filtered buffer on the instrument's cover slide after focussing and directly before measuring. Data were analysed using the DiscoverMP software (Refeyn).

### In vitro reduced MPO–nucleosome binding assay

Reduced MPO monomers (444 nM) were co-incubated with HeLa mononucleosomes (888 nM) in 500 μl of chromatin remodelling buffer without EDTA for 30 min at room temperature. Then, 0.5 μg of MPO

antibody (DAKO) was added to the samples, which were incubated for a further 30 min followed by the addition of 20 μl of protein A/G beads for 20 min. The samples were then washed four times in chromatin remodelling buffer containing 300 mM salt and the beads were then eluted in reducing LDS sample buffer at 70 °C for 10 min before being subjected to western blotting. Nucleosomes were predigested by DNase I as a control.

### Cryo-EM sample preparation, data acquisition, and processing and model building of chemically reduced native MPO samples

Native MPO (2.4 mg ml⁻¹, ~16 μM, Sigma-Aldrich) was incubated for 6 h at room temperature in 10 mM HEPES pH 7.5, 100 mM NaCl and 50 mM DTT. The sample was next mixed with reconstituted nucleosomes (see above) at final concentrations of 1.9 mg ml⁻¹ MPO (~13 μM MPO dimers or ~26 μM MPO monomers if complete reduction and dimer dissociation is assumed) and 1.1 mg ml⁻¹ nucleosomes (~5.6 μM), respectively, and kept on ice for times between 15 s and 60 min.

At the indicated timepoints, 4 μl sample was applied to glow-discharged UltrAuFoil R1.2/1.3 300 grids (Quantifoil) and plunge-frozen in liquid ethane using a Vitrobot Mark IV (Thermo Fisher Scientific). Cryo-EM data were acquired on a 200 kV Talos Arctica microscope (Thermo Fisher Scientific) equipped with a field emission gun. For each timepoint, around 5,000–6,000 micrograph movies were recorded in counting mode (pixel size 0.68 Å px⁻¹) on a K3 camera (Gatan) at a nominal magnification of 130,000×. 70.3 e⁻ Å⁻² were distributed over 60 frames, and the slit width of the Bioquantum electron filter (Gatan) was set to 15 eV. Details of data acquisitions can be found in Supplementary Figs. 17–20 and Supplementary Table 2.

Data processing routes were similar to the data of non-reduced samples (see above) and all steps were performed in cryoSPARC. Three unique molecular assemblies were obtained: nucleosome bound by one and two MPO monomers, respectively, were present in all four timepoints and could be refined to 2.97 Å and 2.95 Å, respectively, in the 2 min timepoint. Nucleosome bound by one MPO monomer and one MPO dimer was present in sufficient number for reconstruction in all but the 2 min timepoint and was refined to 3.16 Å in the 5 min timepoint, followed by focussed refinements on the MPO dimer and the MPO monomer bound to nucleosome, respectively. All details of data processing can be found in Supplementary Figs. 17–21 and Supplementary Table 2.

### Sample preparation and cryo-ET data acquisition of PMA-stimulated NETs

For cryo-ET experiments, neutrophils were purified as described above. Thereafter, neutrophils were applied to glow-discharged cryo-EM grids (Quantifoil, holey carbon film, R 2/1 200). Before freezing, neutrophils were treated with 100 nM PMA (Sigma-Aldrich) for at least 4 h to induce NET formation. Neutrophils were plunge-frozen (37 °C, 90% humidity, backblotting 10 s, blotting force 10 and drain time 2 s), using a Vitrobot Mark IV (Thermo Fisher Scientific). Data were acquired on a 300 kV Titan Krios G3 (Thermo Fisher Scientific) system equipped with a K2 and Bioquantum energy filter (energy width 15 eV, Gatan). Tomograms were taken at a nominal magnification of 42,000× (pixel size 3.445 Å) in a dose symmetric scheme[57] (tilt range, ±48°; tilt increment, 2°). The total exposure of approximately 100 e⁻ Å⁻² was distributed over 49 micrograph movies (8 frames). Details are provided in Supplementary Figs. 6 and 7.

### Cryo-ET data processing

Movies of neutrophil extracellular traps were preprocessed in Warp[58], including patch motion correction and patch CTF estimation (Supplementary Fig. 7a). After tilt series alignment in IMOD (v.4.11)[59], tomograms were reconstructed at binning 4 (pixel size 13.78 Å px⁻¹) in Warp[58]. For picking, the clustering workflow in TomoTwin was used[38]. First, tomograms were embedded with the pretrained model.

To identify possible protein complexes of interest, the embeddings are projected on a 2D manifold (UMAP) (Supplementary Fig. 7b). By annotating protein complexes in the tomogram, clusters are identified in the UMAP. Here, it is possible to identify only the embeddings of possible targets. To clean the UMAP, the identified clusters were used to recalculate the embeddings and therefore the UMAP. After several rounds of UAMP polishing, three possible clusters, with a total number of 29,242 particles, were identified. After extraction at binning 1 (box size $64^3$, pixel size 3.445 Å px$^{-1}$) with Warp[58], subvolumes were projected in $z$ direction (box size for projection $64 \times 64 \times 32$) for 2D classification in Sphire 1.4 (Supplementary Fig. 7c). Particles of cluster 3 belonging to classes that showed clear nucleosome features were selected (1,548 particles) and finally refined in Relion (v.3.14)[49]. Refinement reached a resolution of 31 Å (Supplementary Fig. 7d).

### Immunofluorescence microscopy of semi-thin CF sputum cryosections

CF sputum samples were fixed in 2% PFA and 0.05% glutaraldehyde, gelatin-embedded and infiltrated with 2.3 M sucrose according to the method described previously[60]. For immunofluorescence analysis, 200 nm semi-thin sections were cut with a diamond knife at −79 °C with a RMC MTX/CRX cryo-ultramicrotome (Boeckeler Instruments) and transferred to glass coverslips. The sections were blocked with a buffer containing normal donkey serum, BSA and fish gelatin, incubated overnight using antibodies against MPO DAKO A0398 (1:2,000), citrullinated H3 (Abcam, ab5103, 1:500) or 3D9 (1 μg ml$^{-1}$). During secondary antibody incubations (Alexa Fluor (Invitrogen) or CF dyes (Sigma-Aldrich) 1:500), the sections were counterstained with DNA dyes Sytox (Invitrogen) and or Hoechst 33258 (Invitrogen). The coverslips were mounted onto glass slides with Mowiol (Carl Roth) and analysed using the Leica Thunder widefield microscopy system.

### Reporting summary

Further information on research design is available in the Nature Portfolio Reporting Summary linked to this article.

### Data availability

The cryo-EM SPA data have been deposited at the EMDB and accompanying molecular models at the PDB under the following accession codes. rMPO–nucleosome complex: EMD-51295 and PDB 9GEN. Free nucleosome (5 min timepoint): EMD-51296 and PDB 9GEO. MPO monomer–nucleosome complex (5 min timepoint): EMD-51297 and PDB 9GEP. MPO dimer–nucleosome complex (5 min timepoint): EMD-51298 (nucleosome-focused map), EMD-51299 (MPO-focused map), EMD-51300 (consensus map), EMD-51301 (composite map) and PDB 9GEQ. MPO dimer–nucleosome complex (intermediate state; 15 s timepoint): EMD-51302 nucleosome-focused map), EMD-51303 (MPO-focused map), EMD-51304 (consensus map), EMD-51305 (composite map) and PDB 9GER. MPO monomer–nucleosome complex (30 min plus SEC): EMD-51306. The cryo-EM data regarding the complexes with DTT-reduced native MPO were deposited under the following accession numbers: nucleosome bound by one MPO monomer (2 min dataset): EMD-52865 and PDB 9IHD. Nucleosome bound by two MPO monomers (2 min dataset): EMD-52866 and PDB 9IHE. Nucleosome bound by one MPO monomer and one MPO dimer (5 min dataset): EMD-52867 (consensus map), EMD-52868 (nucleosome/MPO-monomer-focused map), EMD-52869 (MPO-dimer-focused map), EMD-52870 (composite map) and PDB 9IHF. The starting models for the model building can be found in the PDB under the following accession codes: 1MHL (native MPO dimer); 6AZP (recombinant MPO) and 6R1T (nucleosome). All other data are available from the corresponding authors on reasonable request.

### Code availability

The MATLAB scripts used for analysis of light microscopy data are provided in the GitHub repository (https://github.com/ngimber/NanoNET).

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

**Acknowledgements** We thank J. Schoranzer for supporting super-resolution microscopy data acquisition; V. Brinkmann and C. Goosman for assisting with the processing and staining of the sputum samples; O. Hofnagel for assistance with cryo-EM data acquisition and T. Wagner for assistance with cryo-ET data analysis; A. Musacchio, M. Pesenti and D. Vogt for reconstituted nucleosomes for structural studies; A. Foti for measuring the peroxidase activity of MPO; M. Stahl and J. Tattersall-Wong for their help in providing sputum samples; and S. Ruddle and C. H. Harbort for comments on the final manuscript. This work was funded by the Max Planck Society (to S.R. and A.Z.).

**Author contributions** A.Z. and S.R. supervised the project. A.Z. and G.L.B. and conceived the project. S.R., T.R. and S. Tacke planned the structural part of the project. G.L.B. planned and performed all biochemical assays and analysed results. T.R. planned, performed and analysed all single-particle cryo-EM experiments. T.R. and D.P. acquired single-particle data. S. Tacke, T.R. and G.L.B. planned cryo-ET experiments and prepared samples. S. Tacke acquired cryo-ET data and analysed results. M.W., N.G. and G.L.B. performed, acquired and analysed the super-resolution datasets. S. Thee provided sputum samples that were processed by G.L.B.; A.Z., G.L.B., T.R., S.R. and S. Tacke conceptualized the data. T.R. G.L.B. and S. Tacke wrote the manuscript with input from S.R. and A.Z.

**Funding** Open access funding provided by Max Planck Society.

**Competing interests** The authors declare no competing interests.

**Additional information**
**Correspondence and requests for materials** should be addressed to Stefan Raunser or Arturo Zychlinsky.

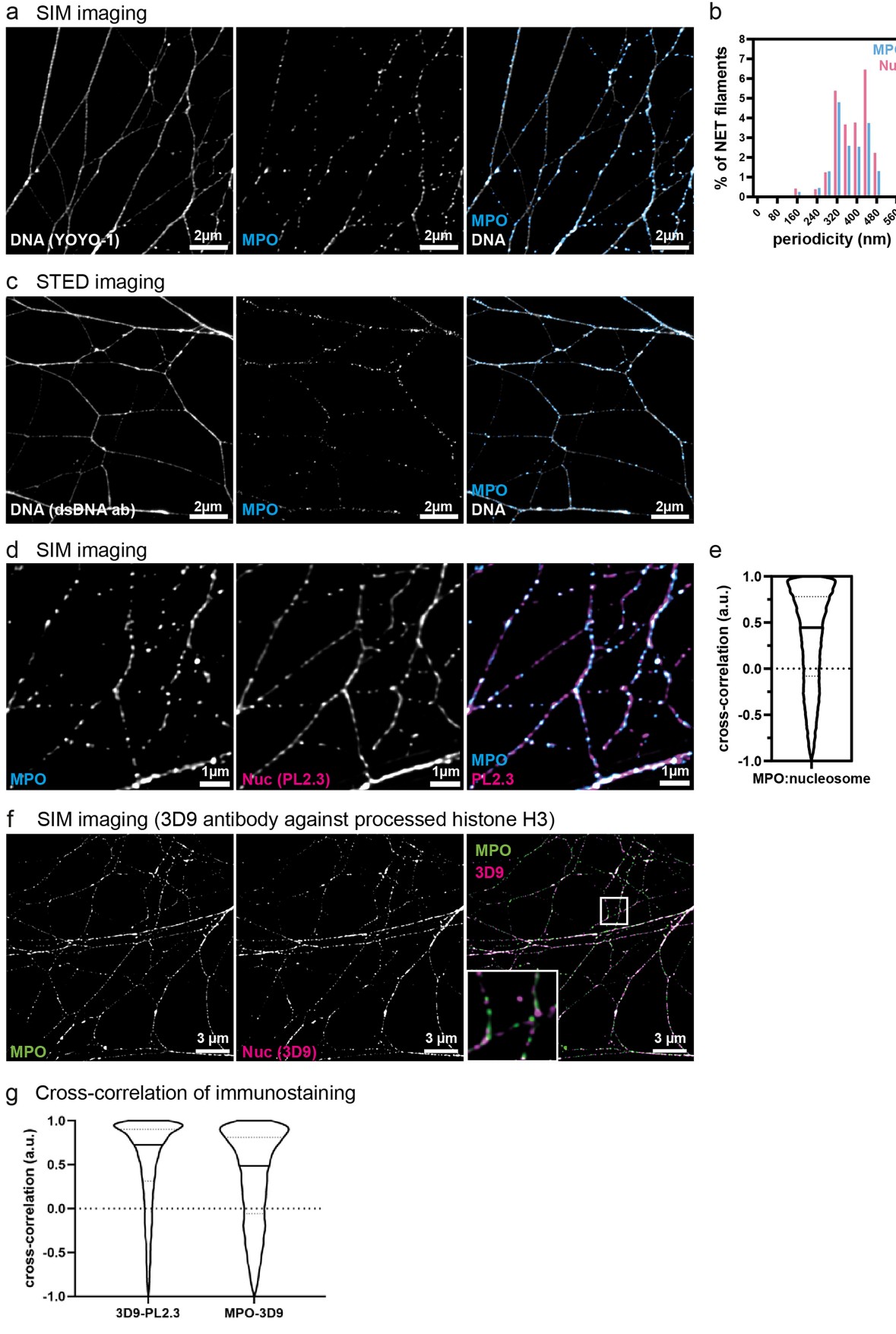

**Extended Data Fig. 1** | See next page for caption.

**Extended Data Fig. 1 | Super-resolution microscopy of NET filaments.**
**a**, Representative SIM image of NETs stained with anti-MPO and DNA dye YOYO-1. $n$ = 5 independent experiments. **b**, Single colour immunofluorescence and autocorrelation of MPO or nucleosomes on the thinnest NET DNA filaments acquired using SIM. $n$ = 5 independent MPO and nucleosome autocorrelation experiments (71,992 NET filament fragments analysed for MPO and nucleosome). **c**, Representative STED image of NETs stained with anti-MPO and anti-DNA. $n$ = 3 independent experiments. **d**, Representative dual colour immunofluorescence of MPO and nucleosome on NETs acquired by SIM. $n$ = 5 independent experiments. **e**, Cross-correlation of MPO and nucleosome quantified from panel **d**. $n$ = 5 independent experiments (71,992 NET filament fragments analysed). **f**, MPO and 3D9 co-localize in SIM microscopy. Neutrophils were stimulated with PMA for 4 h and labelled with antibodies to MPO (green) and 3D9 (magenta), an antibody that recognizes an epitope generated by the proteolytic processing of histone H3 and is NET specific. These data reproduce previous staining with the nucleosomal marker (PL2.3). Scale bar = 3 μm, boxes 2 × 2 μm. $n$ = 4 independent experiments. **g**, MPO colocalises with cleaved histone H3. Cross-correlation (same method as in Fig. 1f) of 3D9 and PL-2.3 demonstrating that these two antibodies are highly colocalized and of MPO with 3D9 demonstrating colocalization as well. $n$ = 4 independent experiments.

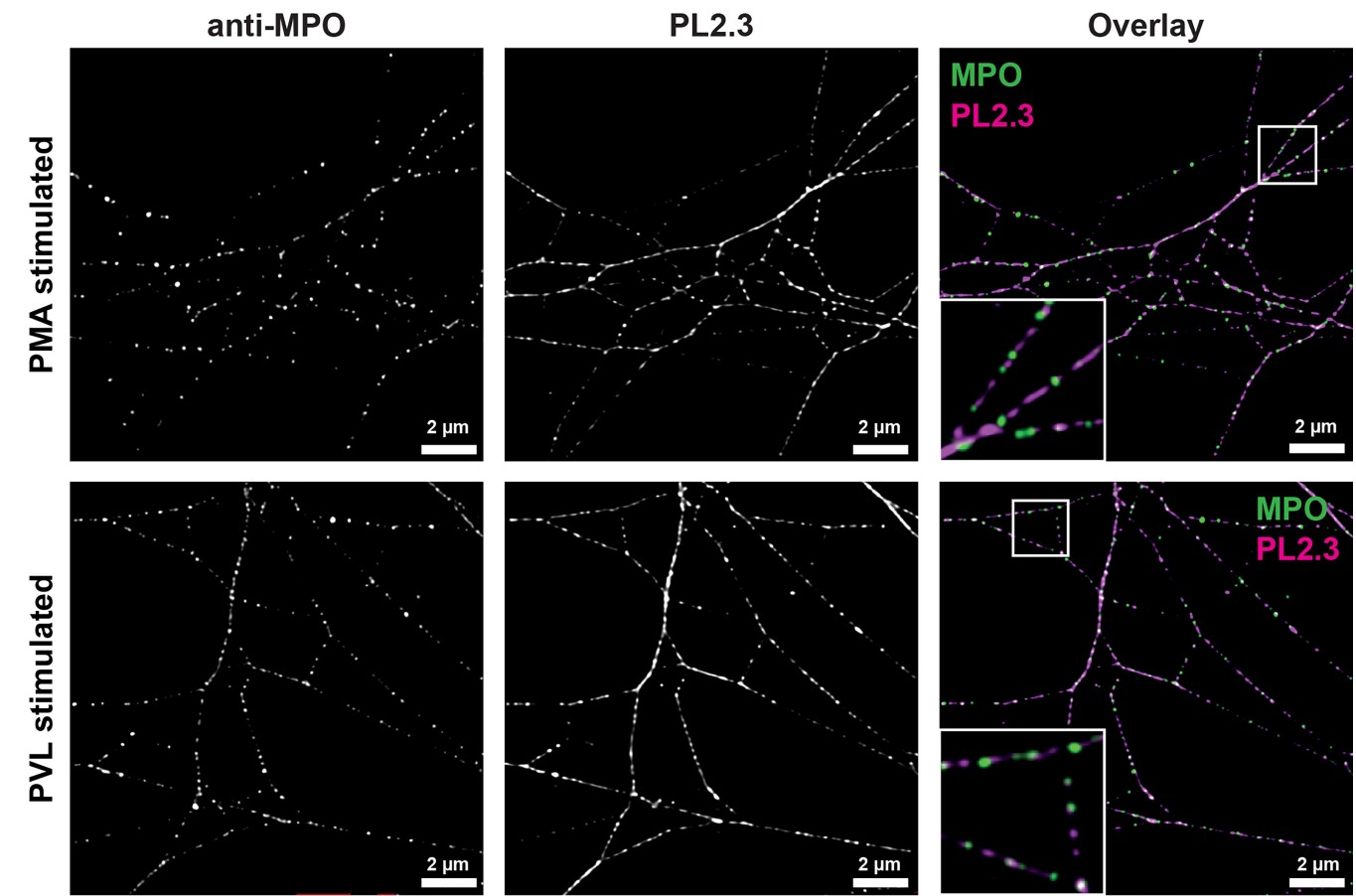

**Extended Data Fig. 2 | MPO shows similar periodicity with different NET stimuli.** Neutrophils were stimulated with PMA or Panton Valentine Leukotoxin (PVL) and stained for MPO (green) and nucleosomes (PL2.3, magenta). Both labels are periodic and are similarly distributed in both stimuli. Boxed zoom-ins show co-localization of MPO and nucleosomes. Scale bar: 2 μm, boxes 2 × 2 μm. *n* = 3 independent experiments.

## a  IP from MSU crystal-induced NETs

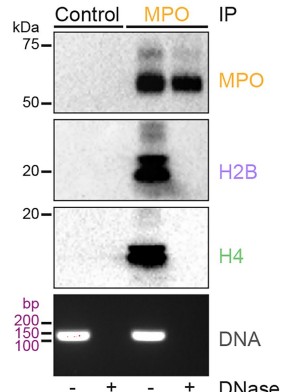

## b  Gel shift assay using rMPO

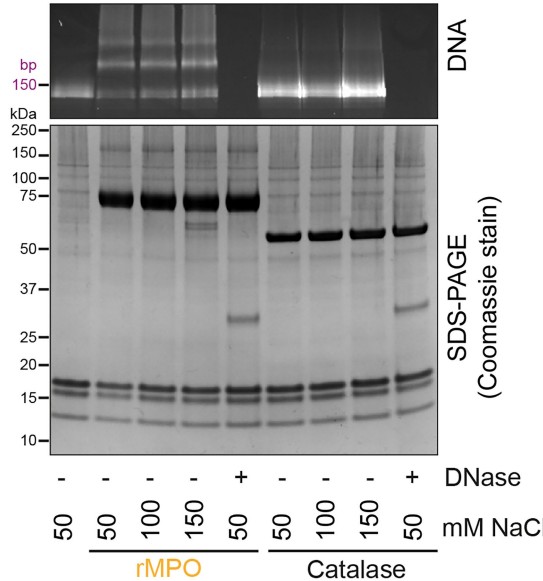

## c  Gel shift assay using MPO

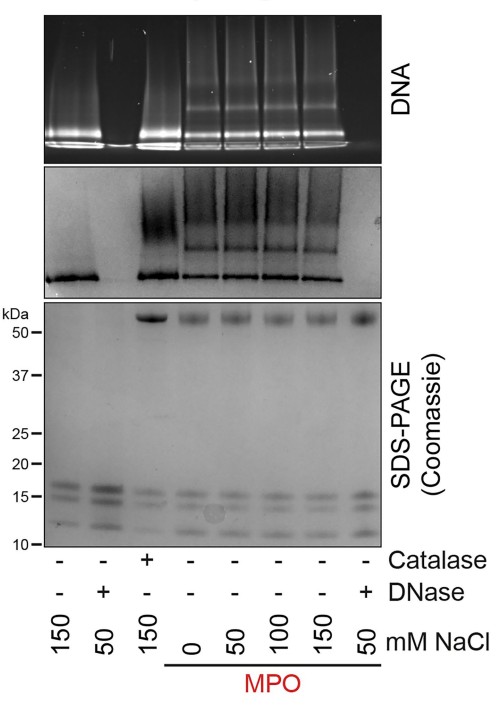

## d  Mass photometry of rMPO and MPO samples

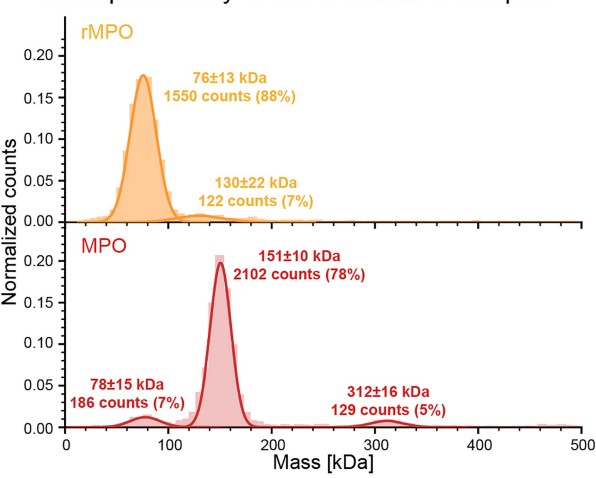

## e  MPO enzymatic activity assay

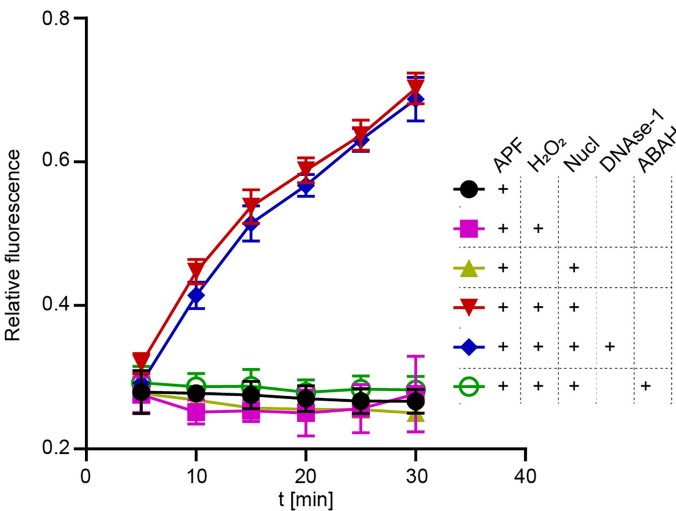

**Extended Data Fig. 3** | See next page for caption.

**Extended Data Fig. 3 | Biochemical characterization of rMPO and MPO preparations. a**, Co-immunoprecipitation of MPO/nucleosome complexes from MSU crystal-induced NETs. MPO immunoprecipitates were blotted for histones and MPO. DNase-I-digested nucleosomes were used as a control. Inputs were calculated from DNA concentrations of each experimental condition and were monitored by agarose gel DNA electrophoresis (bottom panel). $n = 3$ independent experiments using MSU crystals as a NET stimulus. **b**, Native gel shift assay using Hela derived nucleosomes and rMPO or catalase at 50–150 mM NaCl concentrations. Nucleosome shifting was monitored by staining gels with ethidium bromide. DNase-I-digested nucleosome was used as a control and abolished shifted bands and acted as a proxy for nucleosome identification. 10 µl from each reaction was subjected to SDS-PAGE to monitor inputs. $n = 5$ independent experiments. **c**, Native gel shift assay using HeLa derived nucleosomes and native MPO or catalase at 0–150 mM NaCl concentrations. Native gels were stained with ethidium bromide or InstaBlue to monitor band shifts after 10 min of incubation at room temperature. DNase I was used as a control. A small aliquot of each 10 min reaction was monitored using SDS-PAGE. $n = 3$ independent experiments. Uncropped gels and blots of panels **a-c** are shown in Supplementary Fig. 26. **d**, Mass photometry measurements and quantification of purified rMPO and MPO samples at 10 nM final concentration. **e**, Representative example of enzymatic activity of MPO bound versus unbound mononucleosomes. NETs were digested into mononucleosomes using micrococcal nuclease, fractionated and then harvested for enzymatic activity in the presence or absence of DNase I. Error bars represent one standard deviation.

**a** PL2-6/nucleosome complex (PDB 6DZT)

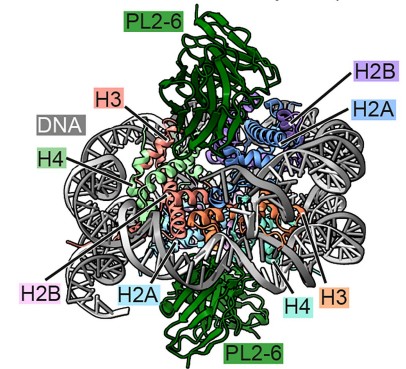

**b** PL2-6 binding to acidic patch (PDB (6DZT)

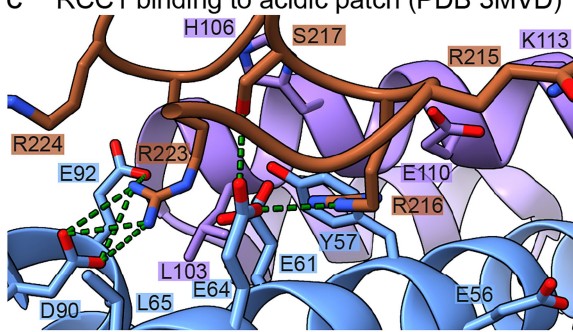

**c** RCC1 binding to acidic patch (PDB 3MVD)

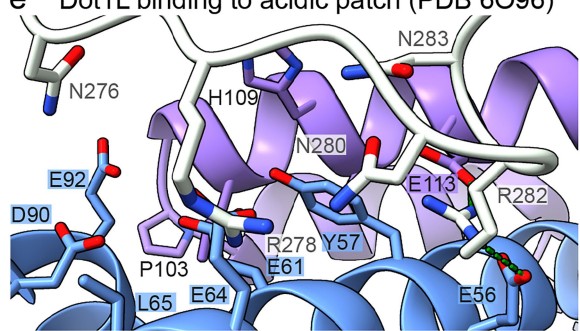

**d** ASir3 BAH domain binding (PDB 3TU4)

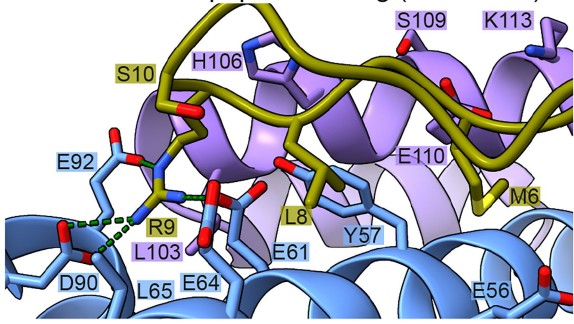

**e** Dot1L binding to acidic patch (PDB 6O96)

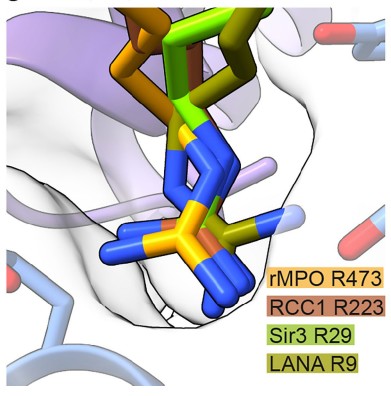

**f** KSHV LANA peptide binding (PDB 1ZLA)

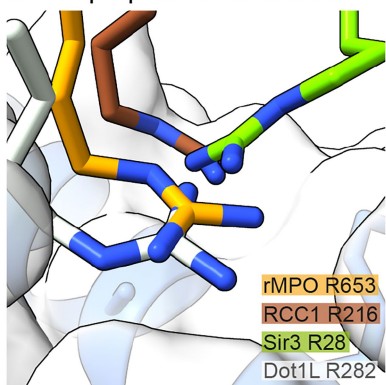

**g** Superposition anchor 1

rMPO R473
RCC1 R223
Sir3 R29
LANA R9

**h** Superposition anchor 2

rMPO R653
RCC1 R216
Sir3 R28
Dot1L R282

**Extended Data Fig. 4** | See next page for caption.

**Extended Data Fig. 4 | Analysis of acidic patch-binding proteins.**
**a**, Structure of the single-chain antibody fragment PL2-6 bound to a nucleosome (PDB 6DZT)[37]. **b**, Close-up of PL2-6 binding to the acidic patch by using two arginine anchors (R124 and R126) that bind in similar positions as the arginine anchors of MPO (Fig. 2d). **c**, Acidic patch binding interface of the chromatin-associated protein RCC1 which is essential to recruit Ran to chromatin (PDB 3MVD)[61]. RCC1 (brown) uses two arginine anchors (R223 and R216) that bind in similar positions as the arginine anchors of MPO (Fig. 2d). **d**, Acidic patch binding interface of the BAH domain of Sir3 which is part of the chromatin silencing complex SIR (PDB 3TU4)[30]. Sir3 (green) uses two arginine anchors (R29 and R28) that bind in similar positions as the arginine anchors of MPO (Fig. 2d). **e**, Acidic patch binding interface of the histone H3 K79 methyltransferase Dot1L (PDB 6O96)[31]. Dot1L (white) uses one arginine anchor (R282) binding to a position similar to MPO arginine anchor 2 (Fig. 2d,f).
**f**, Acidic patch binding interface of the LANA peptide of Kaposi's sarcoma herpesvirus (KSHV; PDB 1ZLA)[62]. The LANA peptide (olive) which is essential for episome persistence uses one arginine anchor (R9) binding to a position very similar to MPO arginine anchor 1 (Fig. 2d,e). **g**, Superposition of position 1 arginine anchors of MPO (R473, orange), RCC1 (R223, brown), Sir3 (R29, green) and the KSHV LANA peptide (R9, olive). For clarity, only the MPO-bound nucleosome is shown as a semi-transparent surface, cartoons and side chains of selected residues. **h**, Superposition of position 2 arginine anchors of MPO (R653, orange), RCC1 (R216, brown), Sir3 (R26, green) and Dot1L (R282, white). For clarity, only the MPO-bound nucleosome is shown as a semi-transparent surface, cartoons and side chains of selected residues.

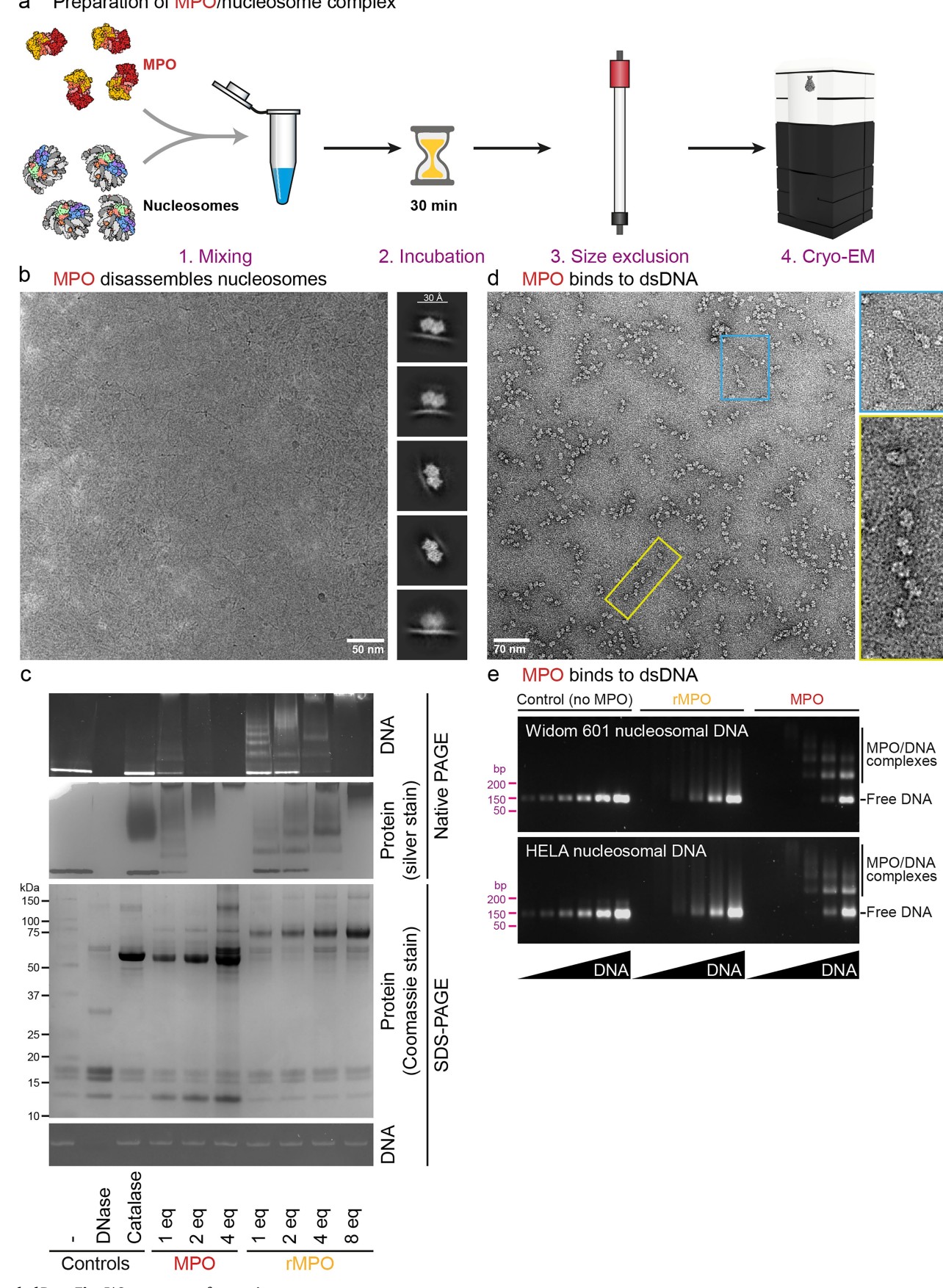

**a** Preparation of MPO/nucleosome complex

MPO

Nucleosomes

1. Mixing 2. Incubation 3. Size exclusion 4. Cryo-EM

30 min

**b** MPO disassembles nucleosomes

30 Å

50 nm

**d** MPO binds to dsDNA

70 nm

**c**

DNA

Protein (silver stain)

Native PAGE

Protein (Coomassie stain)

SDS-PAGE

kDa
150
100
75

50

37

DNA

−  DNase  Catalase  1 eq  2 eq  4 eq  1 eq  2 eq  4 eq  8 eq

Controls MPO rMPO

**e** MPO binds to dsDNA

Control (no MPO) rMPO MPO

Widom 601 nucleosomal DNA

bp
200
150
50

MPO/DNA complexes

Free DNA

HELA nucleosomal DNA

bp
200
150
50

MPO/DNA complexes

Free DNA

DNA DNA DNA

**Extended Data Fig. 5** | See next page for caption.

**Extended Data Fig. 5 | Preparation and analysis of MPO/nucleosome complexes. a**, Scheme showing the reconstitution of complexes comprising native MPO and nucleosomes by mixing (1) and incubation (2), followed by purification using size exclusion chromatography via a Superdex 200 5/150 column (3) and subsequent analysis by cryo-EM (4). **b**, Representative micrograph of a sample prepared via the reconstitution route shown in panel **a**. The sample contains a large amount of free, unbound DNA (thin, elongated filaments) which indicates nucleosome disassembly. Selected 2D classes resemble dimeric MPO bound to a DNA filament. **c**, Native gel shift assay using recombinant nucleosomes and the indicated molar equivalents (eq) of rMPO or MPO, respectively. After 10 min incubation, an aliquot from each reaction was run in native gels, stained with ethidium bromide to visualize DNA and then subsequently silver stained to visualize protein. In addition, reduced SDS-PAGE gels of the same samples were run to visualize total protein and DNA levels in each reaction. $n = 4$ independent experiments. **d**, Representative negative stain micrograph of MPO reconstituted with Widom 601 DNA (without histone proteins). MPO binds to the DNA. **e**, MPO-DNA binding assay. Recombinant or native MPO was incubated with dsDNA (either synthetic Widom-601 DNA or DNA derived from HeLa nucleosomes) and subjected to DNA agarose electrophoresis after 30 min of incubation at room temperature. DNA was visualized by ethidium bromide staining. $n = 3$ independent experiments. Uncropped gels of panels **c,e** are shown in Supplementary Fig. 27.

## a  Time course experiment

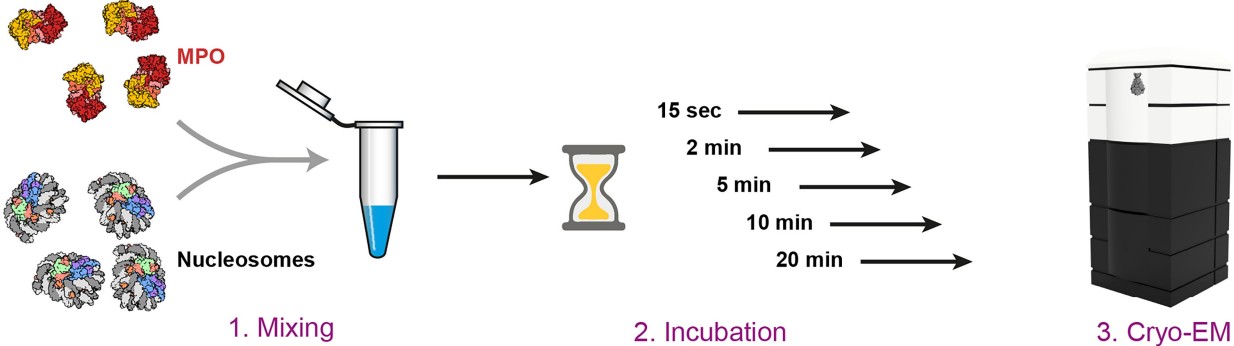

**MPO**

**Nucleosomes**

15 sec
2 min
5 min
10 min
20 min

1. Mixing          2. Incubation          3. Cryo-EM

## b  Cryo-EM reconstructions of 5 minutes time point

**Free nucleosome class**       **MPO monomer/nucleosome class**       **MPO dimer/nucleosome class**

## c  Models derived from reconstructions of 5 minutes time point

**Extended Data Fig. 6 | Preparation and analysis of MPO/nucleosome complexes. a**, Scheme showing the time course experiment for reconstitution of complexes comprising native MPO and nucleosomes. Both components are mixed (1) and incubated for periods between 15 s and 20 min as indicated (2). Then, instead of purifying via size exclusion chromatography, samples were directly subjected to analysis by cryo-EM (3). **b**, Cryo-EM reconstructions derived from the dataset corresponding to the 5 min time point of the time course experiment shown in panel **a**. The dataset is representative also for 2, 10 and 20 min incubations and contains free nucleosomes (left) and MPO monomer/nucleosome (middle) and MPO dimer/nucleosome complexes (right). Details of processing of this dataset are shown in Supplementary Fig. 11 and in Supplementary Table 1. **b**, Molecular models corresponding to the reconstructions shown in panel **b**.

a   Superposition MPO dimer

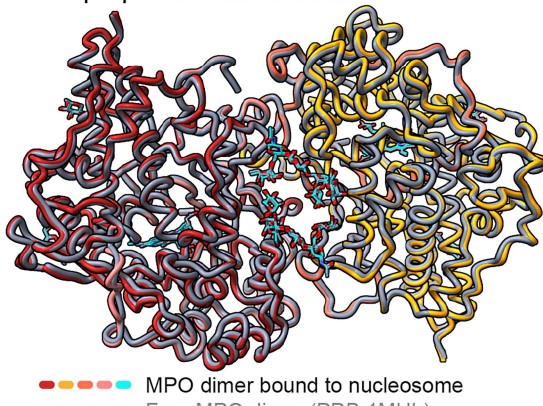

●●●●● MPO dimer bound to nucleosome
━━━ Free MPO dimer (PDB 1MHL)

b   Superposition nucleosome

Nucleosome bound
to MPO dimer
Free nucleosome

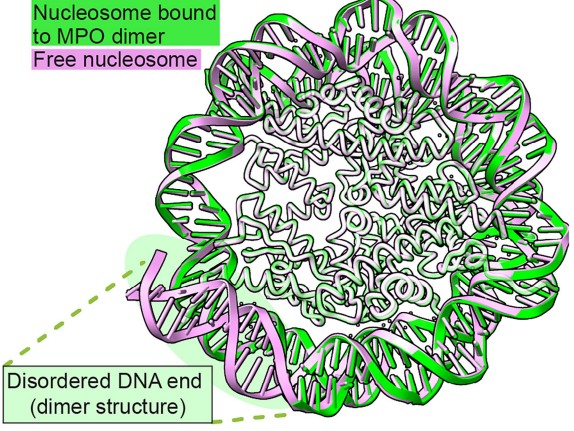

Disordered DNA end
(dimer structure)

c   Superposition nucleosome

Nucleosome bound to:
MPO dimer
rMPO

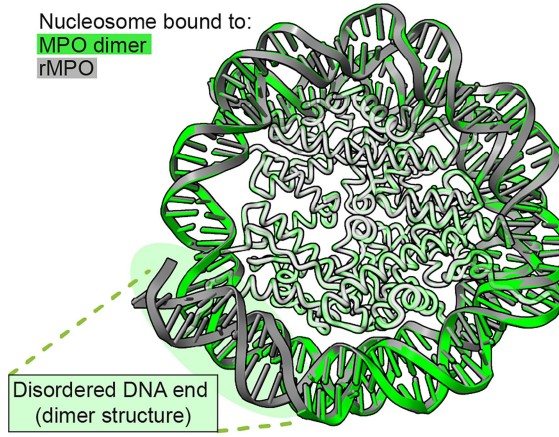

Disordered DNA end
(dimer structure)

d   Superposition nucleosome

Nucleosome bound to:
MPO dimer
MPO monomer

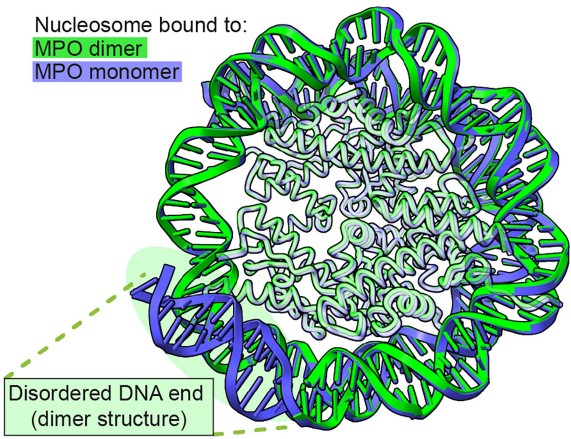

Disordered DNA end
(dimer structure)

**Extended Data Fig. 7 | MPO and nucleosomes in different complexes are structurally almost identical. a**, Superposition of dimeric MPO from the nucleosome-bound cryo-EM structure (colours as in Fig. 3) and in an earlier crystal structure (grey, PDB 1MHL)[55]. Both structures are virtually identical; RMSD 0.44 Å over 1,136 residues. **b**, Superposition of free nucleosome (violet) and the MPO dimer-bound nucleosome (green). Notably, one DNA end (terminal 12 base pairs) of the MPO-bound nucleosome are disordered. RMSD 0.41 Å over 1,016 residues. **c**, Superposition of the rMPO-bound nucleosome (grey) and the MPO dimer-bound nucleosome (green). RMSD 0.50 Å over 1,016 residues. **d**, Superposition of the MPO monomer-bound nucleosome (blue) and the MPO dimer-bound nucleosome (green). RMSD 0.71 Å over 1,016 residues.

**a**  MPO monomer/nucleosome complex (30 minutes plus SEC)

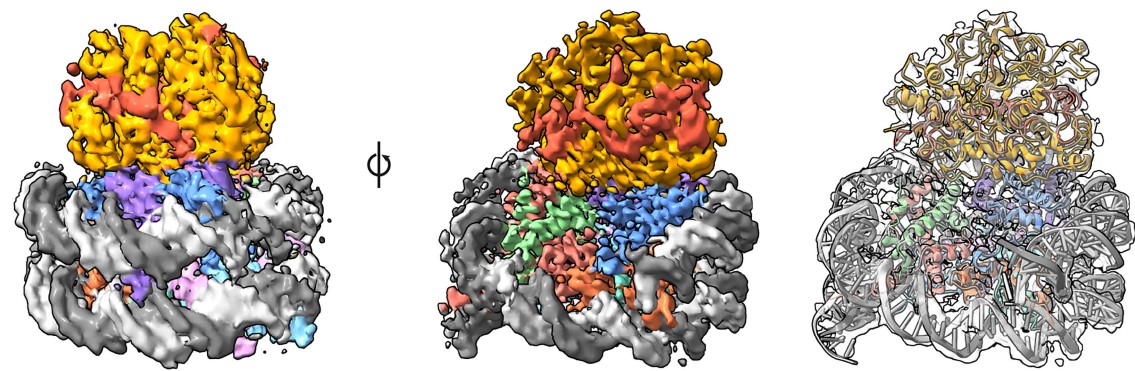

**b**  MPO dimer/nucleosome complex (intermediate conformation; 15 seconds time point)

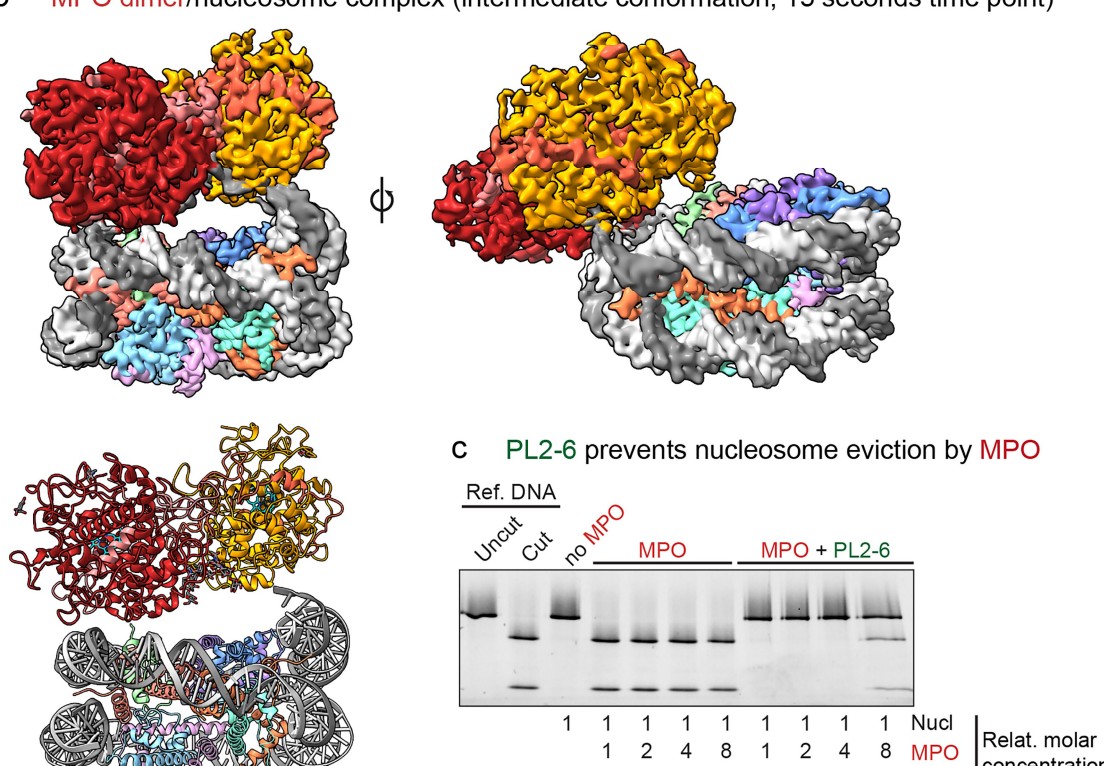

**c**  PL2-6 prevents nucleosome eviction by MPO

| | | | | | | | | | | |
|---|---|---|---|---|---|---|---|---|---|---|
| 1 | 1 | 1 | 1 | 1 | 1 | 1 | 1 | 1 | Nucl | |
| | 1 | 2 | 4 | 8 | 1 | 2 | 4 | 8 | MPO | Relat. molar concentrations |
| | | | | | 3 | 3 | 3 | 3 | PL2-6 | |

**Extended Data Fig. 8 | MPO binds nucleosomes via DNA and acidic patch.**
**a**, Cryo-EM reconstruction (in two orientations) of the MPO monomer/nucleosome complex incubated for 30 min and then purified over SEC. On the right, the reconstruction is shown as a transparent surface with the docked MPO monomer/nucleosome model of the 5 min time point dataset. **b**, Cryo-EM reconstruction (in two orientations) and model of the MPO dimer/nucleosome complex in the intermediate state as observed only in the 15 s time point.

**c**, Nucleosome remodelling assay using recombinant nucleosomes with an encrypted GATC restriction site that is cut by DpnII only when nucleosome-DNA interactions are perturbed. GATC nucleosomes were employed to monitor the kinetics of nucleosome disassembly by MPO in absence and presence of the acidic patch binder PL2-6. $n = 3$ independent experiments. The uncropped gel is shown in Supplementary Fig. 28.

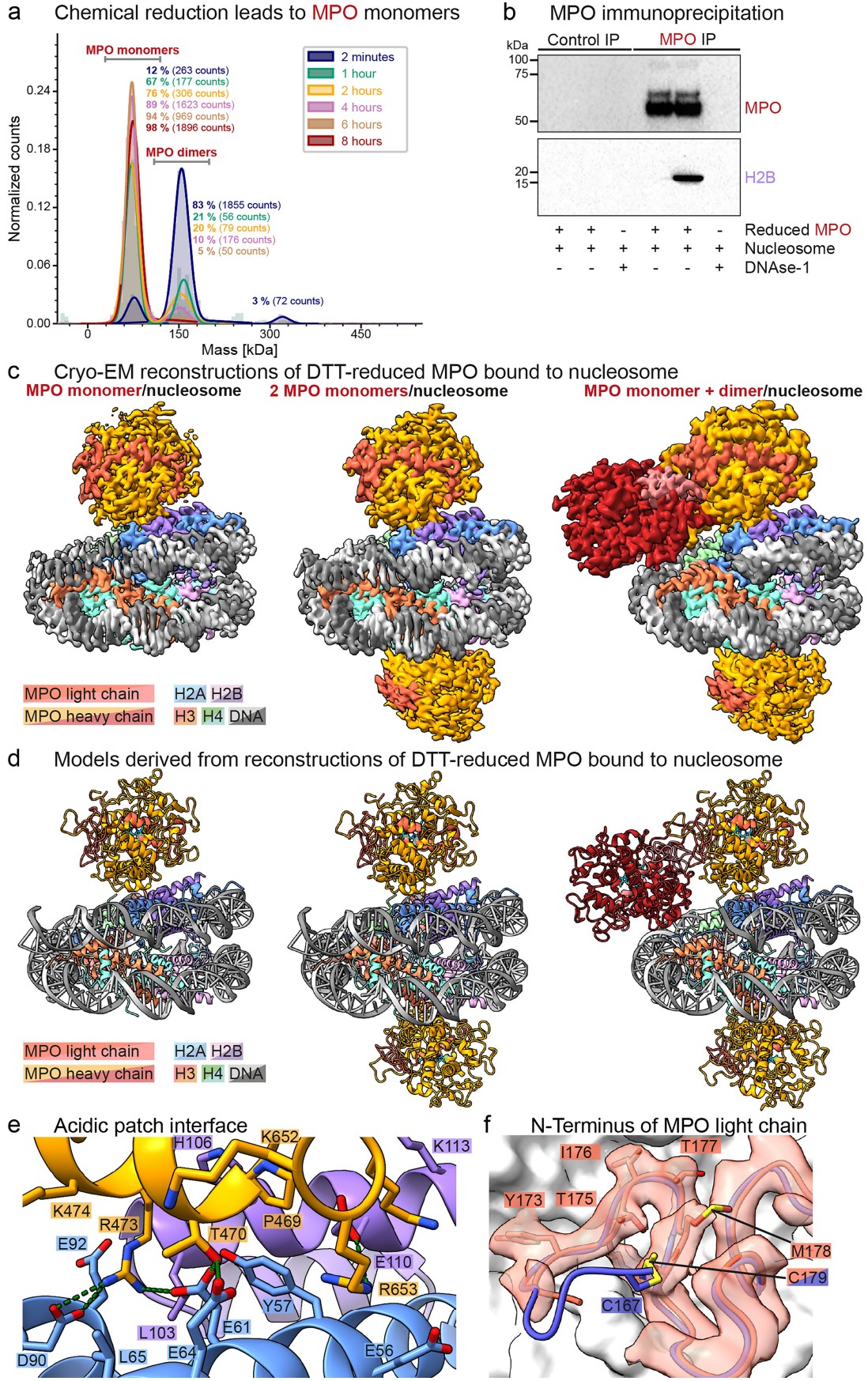

**a** Chemical reduction leads to MPO monomers

**b** MPO immunoprecipitation

**c** Cryo-EM reconstructions of DTT-reduced MPO bound to nucleosome

MPO monomer/nucleosome   2 MPO monomers/nucleosome   MPO monomer + dimer/nucleosome

**d** Models derived from reconstructions of DTT-reduced MPO bound to nucleosome

**e** Acidic patch interface

**f** N-Terminus of MPO light chain

**Extended Data Fig. 9** | See next page for caption.

**Extended Data Fig. 9 | DTT-reduced MPO is capable of binding to acidic patch. a**, Mass photometry of DTT-reduced MPO samples. Incubation of MPO with 50 mM DTT leads to a progressive shift in particle mass from a dominant peak around 150 kDa (MPO dimers) to almost exclusively a peak around 75 kDa (MPO monomer). After 4 h, a similar monomer:dimer ratio as in the (non-reduced) rMPO sample is reached (see Supplementary Fig. 6d. **b**, MPO immunoprecipitation using DTT-reduced MPO and nucleosomes. The observed interaction of MPO with H2B depends on intact nucleosomes as DNase I treatment prevents it. The uncropped blots are shown in Supplementary Fig. 28. **c**, Cryo-EM reconstructions of samples prepared using DTT-reduced MPO and nucleosomes derived from the dataset corresponding to the 2 min (nucleosome bound by one or two MPO monomers, respectively) or 5 min (nucleosome bound by one MPO monomer and one dimer) time points of the time course experiment. Details of processing of this dataset are shown in Supplementary Figs. 18, 19 and in Supplementary Table 1. **d**, Molecular models corresponding to the reconstructions shown in panel **c**. **e**, Acidic patch interface in reduced MPO (structure of nucleosome bound by one MPO monomer). Interaction of MPO with histones H2A (light blue) and H2B (light purple) is very similar as in the case of rMPO or non-reduced MPO. **f**, N-terminus of the MPO light chain gets disordered upon DTT reduction. Before reduction, the N-terminus is fixed by a cystine bridge between C167 and C179 (blue). Upon reduction, the density of the N-terminal four amino acids gets disordered (salmon density), whereas the remaining part of the light chain stays unperturbed. The heavy chain is shown as a white surface. Several side chains of the light chain are shown for reference.

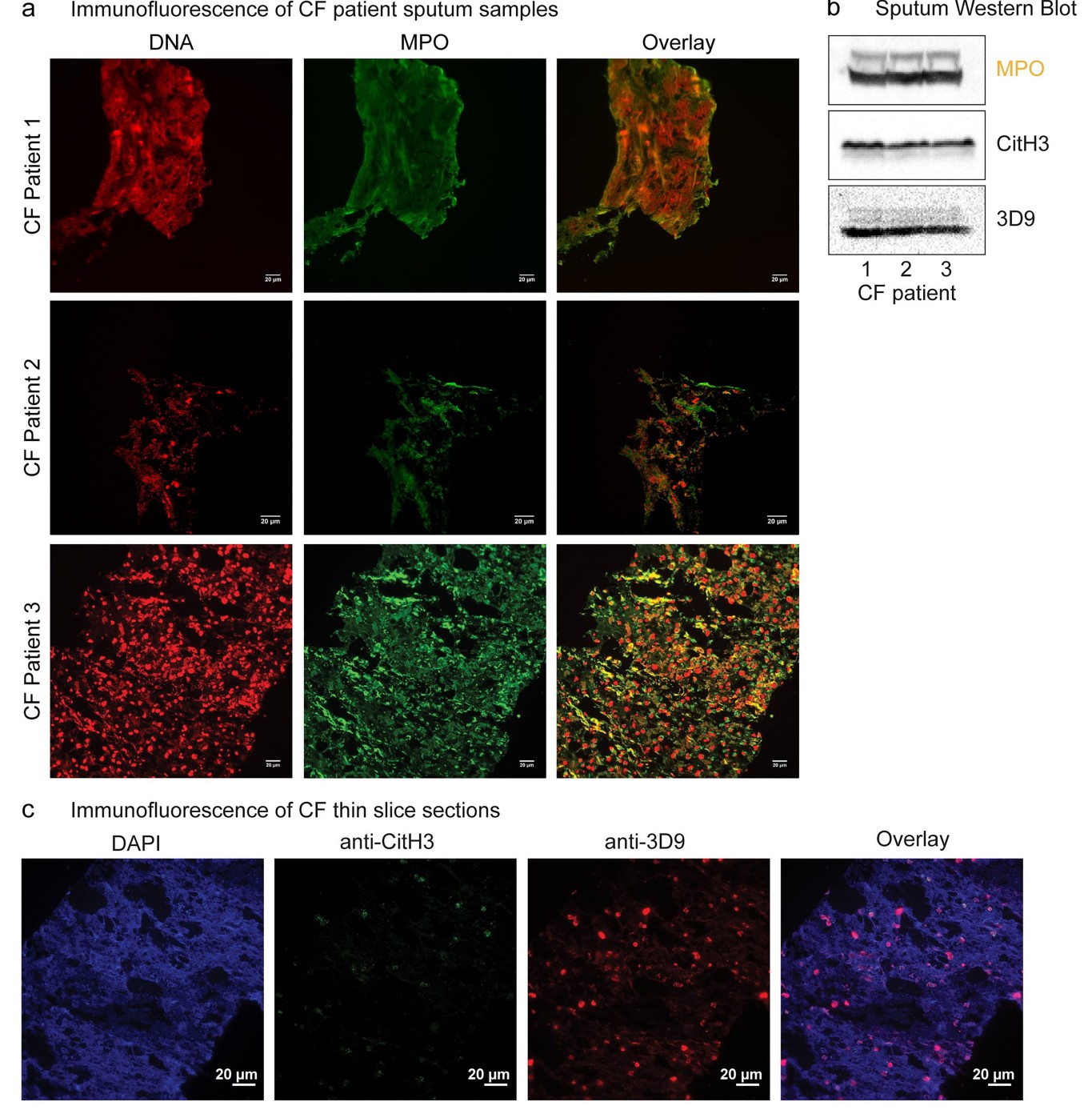

a  Immunofluorescence of CF patient sputum samples

DNA  MPO  Overlay

CF Patient 1

CF Patient 2

CF Patient 3

b  Sputum Western Blot

MPO

CitH3

3D9

1  2  3
CF patient

c  Immunofluorescence of CF thin slice sections

DAPI  anti-CitH3  anti-3D9  Overlay

**Extended Data Fig. 10 | Sputum of cystic fibrosis patients contains MPO-positive NETs. a**, Thin slice cryosections of human CF sputum stained with Sytox to stain DNA (red) and anti-MPO (green). Each CF patient corresponds to the biochemistry performed in Fig. 5c. **b**, Human sputum was digested with DNase I, reduced and then subjected to western blot using antibodies in panel **c**. The uncropped blots are shown in Supplementary Fig. 28. **c**, Thin slice cryosections of human CF sputum stained with antibodies recognizing citrullinated H3 and a neutrophil elastase cleavage site that is also located on the tails of H3. DNA was counterstained with DAPI.

# Reporting Summary

## Statistics

For all statistical analyses, confirm that the following items are present in the figure legend, table legend, main text, or Methods section.

| n/a | Confirmed | |
|---|---|---|
| ☐ | ☒ | The exact sample size (*n*) for each experimental group/condition, given as a discrete number and unit of measurement |
| ☐ | ☒ | A statement on whether measurements were taken from distinct samples or whether the same sample was measured repeatedly |
| ☒ | ☐ | The statistical test(s) used AND whether they are one- or two-sided<br>*Only common tests should be described solely by name; describe more complex techniques in the Methods section.* |
| ☒ | ☐ | A description of all covariates tested |
| ☒ | ☐ | A description of any assumptions or corrections, such as tests of normality and adjustment for multiple comparisons |
| ☒ | ☐ | A full description of the statistical parameters including central tendency (e.g. means) or other basic estimates (e.g. regression coefficient) AND variation (e.g. standard deviation) or associated estimates of uncertainty (e.g. confidence intervals) |
| ☒ | ☐ | For null hypothesis testing, the test statistic (e.g. *F*, *t*, *r*) with confidence intervals, effect sizes, degrees of freedom and *P* value noted<br>*Give P values as exact values whenever suitable.* |
| ☒ | ☐ | For Bayesian analysis, information on the choice of priors and Markov chain Monte Carlo settings |
| ☒ | ☐ | For hierarchical and complex designs, identification of the appropriate level for tests and full reporting of outcomes |
| ☒ | ☐ | Estimates of effect sizes (e.g. Cohen's *d*, Pearson's *r*), indicating how they were calculated |

*Our web collection on statistics for biologists contains articles on many of the points above.*

## Software and code

Policy information about availability of computer code

| Data collection | EPU v2.7<br>EMMENU 4.0.9.55<br>AcquireMP v2024.2.0 |
|---|---|
| Data analysis | CryoSPARC v4.3 and v4.4 (also CryoSPARC live)<br>Phenix v1.20 and v1.21<br>COOT v0.9.8.2<br>UCSF ChimeraX v1.8 and v1.9<br>Warp 1.0.9<br>TomoTwin 0.9.0<br>SPHIRE 1.5<br>IMOD 4.11<br>RELION 3.1.4<br>DiscoverMP v2024.2.0<br>https://github.com/ngimber/NanoNET |

For manuscripts utilizing custom algorithms or software that are central to the research but not yet described in published literature, software must be made available to editors and reviewers. We strongly encourage code deposition in a community repository (e.g. GitHub). See the Nature Portfolio guidelines for submitting code & software for further information.

## Data

Policy information about availability of data

All manuscripts must include a data availability statement. This statement should provide the following information, where applicable:

- Accession codes, unique identifiers, or web links for publicly available datasets
- A description of any restrictions on data availability
- For clinical datasets or third party data, please ensure that the statement adheres to our policy

The cryo-EM SPA data have been deposited at the EMDB and accompanying molecular models at the PDB under the following accession codes. rMPO/nucleosome complex: EMD-51295 (https://www.ebi.ac.uk/emdb/EMD-51295) and PDB 9GEN (https://doi.org/10.2210/pdb9GEN/pdb). Free nucleosome (5 minutes time point): EMD-51296 (https://www.ebi.ac.uk/emdb/EMD-51296) and PDB 9GEO (https://doi.org/10.2210/pdb9GEO/pdb). MPO monomer/nucleosome complex (5 minutes time point): EMD-51297 (https://www.ebi.ac.uk/emdb/EMD-51297) and PDB 9GEP (https://doi.org/10.2210/pdb9GEP/pdb). MPO dimer/nucleosome complex (5 minutes time point): EMD-51298 (https://www.ebi.ac.uk/emdb/EMD-51298; nucleosome focused map), EMD-51299 (https://www.ebi.ac.uk/emdb/EMD-51299; MPO focused map), EMD-51300 (https://www.ebi.ac.uk/emdb/EMD-51300; consensus map), EMD-51301 (https://www.ebi.ac.uk/emdb/EMD-51301; composite map) and PDB 9GEQ (https://doi.org/10.2210/pdb9GEQ/pdb). MPO dimer/nucleosome complex (intermediate state; 15 seconds time point): EMD-51302 (https://www.ebi.ac.uk/emdb/EMD-51302; nucleosome focused map), EMD-51303 (https://www.ebi.ac.uk/emdb/EMD-51303; MPO focused map), EMD-51304 (https://www.ebi.ac.uk/emdb/EMD-51304; consensus map) EMD-51305 (https://www.ebi.ac.uk/emdb/EMD-51305; composite map) and PDB 9GER (https://doi.org/10.2210/pdb9GER/pdb). MPO monomer/nucleosome complex (30 minutes plus SEC): EMD-51306 (https://www.ebi.ac.uk/emdb/EMD-51306). The cryo-EM data regarding the complexes with DTT-reduced native MPO were deposited under the following accession numbers: Nucleosome bound by one MPO monomer (2 minutes dataset): EMD-52865 (https://www.ebi.ac.uk/emdb/EMD-52865) and PDB 9IHD (https://doi.org/10.2210/pdb9IHD/pdb). Nucleosome bound by two MPO monomers (2 minutes dataset): EMD-52866 (https://www.ebi.ac.uk/emdb/EMD-52866) and PDB 9IHE (https://doi.org/10.2210/pdb9IHE/pdb). Nucleosome bound by one MPO monomer and one MPO dimer (5 minutes dataset): EMD-52867 (https://www.ebi.ac.uk/emdb/EMD-52867; consensus map), EMD-52868 (https://www.ebi.ac.uk/emdb/EMD-52868; nucleosome/MPO monomer focused map), EMD-52869 (https://www.ebi.ac.uk/emdb/EMD-52869; MPO dimer focused map), EMD-52870 (https://www.ebi.ac.uk/emdb/EMD-52870; composite map) and PDB 9IHF (https://doi.org/10.2210/pdb9IHF/pdb). The starting models for the model building can be found in the PDB under the following accession codes: 1MHL (https://doi.org/10.2210/pdb1MHL/pdb; native MPO dimer); 6AZP (https://doi.org/10.2210/pdb6AZP/pdb; recombinant MPO) and 6R1T (https://doi.org/10.2210/pdb6R1T/pdb; nucleosome). The utilized MatLab scripts for analysis of light microscopy data can be found in the GitHub repository (https://github.com/ngimber/NanoNET). All other data are available from the corresponding authors upon reasonable request.

## Research involving human participants, their data, or biological material

Policy information about studies with human participants or human data. See also policy information about sex, gender (identity/presentation), and sexual orientation and race, ethnicity and racism.

| | |
|---|---|
| Reporting on sex and gender | This information was not collected |
| Reporting on race, ethnicity, or other socially relevant groupings | N/A |
| Population characteristics | No data on population characteristics were collected |
| Recruitment | Anonymous blood donations from the Charité Campus Campus Mitte blood bank were purchased. Sputum samples were collected by Dr. Stephanie Thee after explicit consent was obtained where patients could opt out. |
| Ethics oversight | Sample collection were approved by the ethics committee of Charité University Hospital, Berlin, Germany |

Note that full information on the approval of the study protocol must also be provided in the manuscript.

# Field-specific reporting

Please select the one below that is the best fit for your research. If you are not sure, read the appropriate sections before making your selection.

☒ Life sciences ☐ Behavioural & social sciences ☐ Ecological, evolutionary & environmental sciences

For a reference copy of the document with all sections, see nature.com/documents/nr-reporting-summary-flat.pdf

# Life sciences study design

All studies must disclose on these points even when the disclosure is negative.

| | |
|---|---|
| Sample size | The choice of sample size (at least three independent experiments) was based on common practice in the field. |
| Data exclusions | No data were excluded |
| Replication | All data were faithfully replicated by performing at least 3 independent experiments or more replications, as indicated in figure legends. |

| Randomization | This is not relevant to our study as experiments were performed in solo and therefore it was impossible to randomise the samples |
|---|---|
| Blinding | Blinding was not possible due to the fact that patient samples were collected when available on the day. All blood bank samples were anonymous. |

# Reporting for specific materials, systems and methods

We require information from authors about some types of materials, experimental systems and methods used in many studies. Here, indicate whether each material, system or method listed is relevant to your study. If you are not sure if a list item applies to your research, read the appropriate section before selecting a response.

## Materials & experimental systems

| n/a | Involved in the study |
|---|---|
| ☐ | ☒ Antibodies |
| ☐ | ☒ Eukaryotic cell lines |
| ☒ | ☐ Palaeontology and archaeology |
| ☒ | ☐ Animals and other organisms |
| ☒ | ☐ Clinical data |
| ☒ | ☐ Dual use research of concern |
| ☒ | ☐ Plants |

## Methods

| n/a | Involved in the study |
|---|---|
| ☒ | ☐ ChIP-seq |
| ☒ | ☐ Flow cytometry |
| ☒ | ☐ MRI-based neuroimaging |

## Antibodies

| Antibodies used | Dako rabbit anti-myeloperoxidase cat number A0398, lots 41244899/20051734/20030245<br>Cell signal mouse anti-H2A clone number L88A6 catalogue number 36365, lot 1<br>Abcam rabbit anti-H2B (ab1790) lot GR3267844-1<br>Abcam rabbit anti-H3 (ab1791) lot GR64776-1<br>Abcam rabbit anti-H4 (ab10158) lot GR3186358-1<br>anti-CitH3 (ab5103)<br>3D9 (in house generated) |
|---|---|
| Validation | The commercial anti-histone antibodies have been validated and quality tested by the manufacturer for the methods we have applied them to. Details can be found on their websites:<br><br>anti-H2A: https://www.cellsignal.com/products/primary-antibodies/histone-h2a-l88a6-mouse-mab/3636<br>anti-H2B: https://www.abcam.com/en-us/products/primary-antibodies/histone-h2b-antibody-chip-grade-ab1790<br>anti-H3: https://www.abcam.com/en-us/products/primary-antibodies/histone-h3-antibody-nuclear-marker-and-chip-grade-ab1791<br>anti-H4: https://www.abcam.com/en-us/products/primary-antibodies/histone-h4-antibody-chip-grade-ab10158<br>anti-CitH3: https://www.abcam.com/en-us/products/primary-antibodies/histone-h3-citrulline-r2-r8-r17-antibody-ab5103<br><br>The anti-MPO antibody was validated in neutrophil like PLB-985 MPO knockout cells.<br><br>3D9 and PL2.3 were validated in the following references (also cited in the manuscript):<br>Losman, M. J., Fasy, T. M., Novick, K. E. & Monestier, M. Monoclonal autoantibodies to subnucleosomes from a MRL/Mp(-)+/+ mouse. Oligoclonality of the antibody response and recognition of a determinant composed of histones H2A, H2B, and DNA. J. Immunol. 148, 1561–1569 (1992)<br>Dorothea Ogmore Tilley et al. Histone H3 clipping is a novel signature of human neutrophil extracellular traps eLife 11:e68283 (2022) |

## Eukaryotic cell lines

Policy information about cell lines and Sex and Gender in Research

| Cell line source(s) | CVCL_2162 (PLB-985). Sex of cell, Female. Age at sampling, 36Y. Category, Cancer cell line. Provided by Mary Dinauer lab |
|---|---|
| Authentication | not authenticated, only used for antibody verification |
| Mycoplasma contamination | The PLB-985 cell line was used exclusively as a reference or control, and mycoplasma testing of the cell lines was not performed. |
| Commonly misidentified lines<br>(See ICLAC register) | No commonly misidentified cell lines were used in this study |

## Plants

Seed stocks
N/A

Novel plant genotypes
N/A

Authentication
N/A

