## [Peer Review File · Nature]

Myeloperoxidase transforms chromatin into Neutrophil Extracellular Traps

Corresponding Author: Professor Arturo Zychlinsky

Version 0:

Reviewer comments:

Referee #1

(Remarks to the Author)

This paper resolves the elusive issue of how MPO can both participate in NET formation, and bind to NETs themselves, depending on its oligomeric status. The biological implications are novel and provide the first description (and a very detailed one) of how some of the numerous proteins bound to NETs interact with these chromatin scaffolds.

The study is well thought out, well executed, and very innovative. However, there remains aspects that should absolutely be addressed, as listed below.

Major points

1. In the Introduction (line 81), the requirement of MPO is presented as if it were a general feature of NETosis. The authors quoted one of their previous studies which showed that this is indeed the case for two stimuli (PMA and *C. albicans*), and that no NETs are made in individuals fully deficient in MPO in response to the same stimuli. However, a subsequent study from the authors' lab (Kenny et al. 2017, PMID 28574339) reported that while neutrophils from a MPO-deficient patient did NET in response to nigericin or a calcium ionophore. Therefore, it would be more accurate to state that MPO is required *for some* NET stimuli and that accordingly, MPO-deficient individuals cannot form NETs *in response to those stimuli, but can do so in response to other neutrophil activators* (quote the Kenny et al. study).
2. In the submitted manuscript, the authors used two NETosis stimuli, a chemical mitogen (PMA) and a potassium ionophore from an African soil fungus (nigericin), to convincingly demonstrate that they promote an association of MPO with NETs, where MPO is distributed discontinuously, co-localizing with nucleosomes. However, both stimuli are ROS inducers, whereas there are numerous reports in the literature showing that NETosis can be achieved using stimuli that are ROS-independent. Thus, the question remains, i.e. would MPO associate with nucleosomes even when physiological, ROS-independent stimuli are used? (e.g., TNF, G-/GM-CSF, IL-8, MSU crystals, etc) In other words, is the phenomenon described in the manuscript a general feature of NETosis, or is it restricted to some stimuli? (perhaps ROS-inducing ones?) Both possible answers to this fundamental question are highly interesting and pertinent, and as a result, would greatly benefit from being clarified experimentally – especially since this study has the potential to be widely cited.
3. The finding, that complexes containing MPO, various histones and DNA can be immunoprecipitated from either human CF sputum samples or in vitro-generated NETs from PMA-activated neutrophils (Fig 5C), does show that MPO can interact with nucleosomes in vivo, as carefully worded by the authors. But it also suggests something far more interesting, i.e. that there could be an in vivo equivalent to the phenomenon characterized in vitro by the authors. Because there is much cell death in CF airways, resulting in necrotic DNA containing histones, and given that MPO can associate with DNA, the presence of complexes containing all of these components could very well arise independently of whether NETs were also initially present. A rather straightforward experimental demonstration that at least some of these complexes did stem from NETs would be to blot the immunoprecipitates for CitH3 and/or CitH4. It is in fact quite surprising that the authors did not investigate this possibility, especially since it would greatly improve the in vivo relevance of the reported findings (if conclusive).

4. A positive outcome of the above experiments would allow for a very interesting addition to the Discussion, namely that MPO binding to nucleosomes is not only a reflection of its presence in the same extracellular microenvironment as NETs, but probably a central mechanism of chromatin decondensation during NETosis. This alone would considerably increase the biological impact of the findings. But whatever the outcome, the issue of whether the MPO mechanism described herein is a general feature (or not) of NETosis should absolutely be discussed because it has inevitable implications when considering future therapeutic interventions.

Minor points

These collectively revolve around precisions that would very much deserve to be made.

- Line 47: “membrane bound granules” is somewhat misleading, as it could be understood to mean that said granules are bound to the plasma membrane. Perhaps changing for “membrane-enclosed” would be closer to what the authors intended.
- Line 79: it should be specified that NET-bound MPO can kill micro-organisms when H₂O₂ is also present. This would be a more faithful reflection of what was reported in reference 20.
- Line 314: the sentence should read “can initiate” or end with “under some circumstances” since it is not currently known whether this is a general feature of NETosis, or whether it is one of the ways in which NETosis can take place but is otherwise dispensable.

Referee #2

(Remarks to the Author)

Review of Manuscript by Burn et al.:

The manuscript by Burn et al. provides a thorough and insightful exploration of how Myeloperoxidase (MPO) remodels extracellular DNA from neutrophils (NETs), focusing on DNA fragments that are partially encapsulated in nucleosomes. Using a sophisticated series of cellular, biochemical, and structural experiments, the authors demonstrate that monomeric MPO (modelled here with recombinant MPO) forms complexes with nucleosomes. The canonical binding mode observed is driven by interaction with the nucleosomal acidic patch. Interestingly, MPO also appears in a dimeric complex, where one MPO molecule binds canonically while the second molecule engages directly with DNA. This configuration is proposed to promote nucleosome dissociation, primarily due to a clash between MPO and DNA. Overall, this is a well-executed study that significantly enhances our understanding of how MPO, a prominent protein in neutrophils, interacts with extracellular DNA. However, I have major reservations about the authors' main conclusion that MPO acts as a “transformer” of DNA on nucleosomes and that the dimeric MPO form promotes DNA dissociation from nucleosomes.

Major Points:

1. Interpreting rMPO vs. MPO:

Both recombinant MPO (rMPO) and native MPO appear effective at DNA dissociation, with native MPO potentially more efficient (Fig. 3A). The differences are interpreted in light of monomeric behaviour (rMPO) and dimeric behaviour of the native form, though I could not find the biochemical data supporting this. A potential compounding factor could be lower specific activity of rMPO for DNA binding (and/or nucleosome binding). Does the rMPO have the same nonspecific affinity to DNA as native MPO? Perhaps the lack of maturation of rMPO leads to lower DNA binding specificity, lower nucleosome binding affinity, and lower enzymatic activity, and hence indirectly impact its proposed “transformer” activity. This should be measured to confirm. I would further suggest expressing the mature form of MPO recombinantly by co-expressing heavy and light chains (excluding residues 107–112). Introducing mutations based on the dimer structures could render MPO a monomer, and these modified proteins could then be tested for DNA and nucleosome binding and activity. Since rMPO serves as a key monomeric control throughout the manuscript (Fig. 1I, 3A, SFig. 5, etc.), a more refined monomeric MPO sample would be beneficial throughout. A minor question for Fig. 3A also concerns the nature of the biotinylated nucleosomes from Epicypher, which remains unclear in the methods section.

2. Interpretation of DNA Dissociation from Nucleosomes by Native MPO Binding (p.7, line 186):

My primary concern involves the interpretation that native MPO binding induces DNA dissociation from nucleosomes. DNA dissociation is commonly observed *in vitro* when nucleosomes interact with DNA-binding proteins (e.g., PMID: 30250250). With the authors' definition, many of these could be termed “chromatin transformers.” Upon protein binding to nucleosomes, DNA also often dissociates at entry/exit sites in cryo-EM experiments in the absence of steric clashes (e.g., PMID: 29323273, PMID: 29626188). In light of published nucleosome/complex structures, the MPO-nucleosome structure and the DNA clash here does not represent a new class of nucleosome binders, in my opinion.

The observed binding compatibility between dimeric MPO and nucleosomes, as shown in Fig. 3, undermines the proposed incompatibility. This complex is after all stable enough for imaging, at least following a short incubation (and even without crosslinking)! It is possible that more native nucleosomes, other than 601, could promote facilitated DNA displacement. However, this is by no means a given, as native nucleosomes could also better accommodate the DNA displacements needed to bind dimeric MPO. In some endogenous (and Widom 601) structures, unwrapping of 20–30 bp or more is observed (e.g., PMID: 38267599; PMID: 32327602). In my opinion, the incompatibility between MPO and nucleosomes is unnecessarily over-interpreted throughout the manuscript. There may simply be multiple binding modes (e.g., MPO bound as a monomer to nucleosomes, as a dimer, and as a dimer/oligomer to DNA). Overall, I also do not find cryo-EM to be a good method for quantifying DNA-nucleosome stability. If claimed, this is best done at the single-molecule level.

3. Figure 4D – DNA Release Dynamics vs. Nucleosome Disassembly:

This assay assesses DNA release dynamics rather than complete nucleosome disassembly. Temporary DNA release may be required for enzymatic cutting without complete dissociation. Restriction assays are highly sensitive to small perturbations and should be interpreted with caution. Including the position of the DpnII site in the main text would clarify this assay's interpretation. Additionally, as stated previously, a more suitable monomeric MPO control is needed beyond rMPO.

4. Abstract (p.1):

The authors state, "MPO monomers, on the other hand, stably interact with the nucleosome acidic patch without making concomitant DNA contacts, thus explaining how MPO readily binds to and licenses NETs to confer hypohalous acid production in the extracellular space." It is understood that the active site remains unoccluded in these complexes, but peroxidase activity is not explicitly measured. To discuss nucleosome binding's functional impact, such measurements are necessary. Since nucleosome binding could allosterically affect MPO activity, either activating or inhibiting it, the authors should test the effect of nucleosome binding on MPO's activity. This data might even prove more relevant to MPO's mechanism than the proposed "transformer" concept.

5. p.10, line 285 – Tomography Experiments:

The tomography experiments are impressive; however, at 31 Å resolution, distinguishing between monomeric and dimeric MPO configurations is challenging (as stated). Higher-resolution data would strengthen interpretations regarding binding states, but I understand that this is not possible.

Minor Points:

6. Map Quality:

The cryo-EM maps look good, but as a best practice, the local resolution of MPO versus nucleosome binding should be reported.

7. Acidic Patch Binding Classification (p.6, line 152):

The classification of acidic patch binders, as described by Tang et al. (PMID: 34198054), should be cited and applied when detailing acidic patch binding.

Referee #3

(Remarks to the Author)

This study from the Zychlinsky lab focuses on the granule-derived protein, myeloperoxidase (MPO), and aims to characterize it as a 'chromatin transformer' (in contrast to a remodeler)—one that binds nucleosomal DNA sequences/nucleosomes to make chromatin non-replicative and non-encoding and enabling its cell-free biotoxic functions. Overall, the study is interesting and the structural studies are a nice start to defining the potential mechanism. However, the authors do not do any mutagenesis to test, evaluate, and extend the validity of their model(s), and I would have expected this for a study considered at Nature.

Specifically, the authors show that rMPO binds nucleosomes at the acidic patch, but also at the "auxiliary interface" between rMPO residues M688, R691, and Q692 with various H3 residues and posit that this may provide rigidity to the overall assembly. Several mutations need to be made and thoroughly assessed in the assays presented, including: (a) mutations in MPO that disrupt nucleosome AP binding, (b) mutations in MPO that alter auxiliary site binding, (c) mutations in the nucleosome AP to show whether binding of MPO is mitigated, (d) mutations in the MPO active site to characterize binding.

Along the lines of the structure, the authors characterize MPO as a 'chromatin transformer', making many comparisons to chromatin remodeling complexes, but the chromatin/transcription fields would argue that complexes such as PRC1, transcription factors such as Sox2 and Sox11, viral proteins, and others have chromatin transformer-like capabilities and question whether a unique term can really be applied to this mechanism (instead of just general, and still interesting, chromatin binding and activity on chromatin substrates).

To determine how physiologic their findings are, the authors should run an MPO binding panel to evaluate whether common histone modifications that would likely be present in cells on NET-associated chromatin impact MPO-nucleosome binding and dimerization. For example, the authors could purify NET-associated chromatin and determine the most abundant modifications/variants to then inform a set of nucleosomes to evaluate. This would strengthen the cellular relevance of their structural findings.

In figure 1, the authors use relatively non-specific inhibitors of MPO (i.e. sodium azide) and conclude that MPO activity is dispensable for MPO-nucleosome association. Can the authors make a point mutation in the catalytic pocket that renders MPO inactive and use this to conclude firmly in parallel?

In Figure 1, the authors show that MPO associates with NCPs in cells and directly in vitro.

Staining is shown for MPO and "Nuc". The authors should please stain for multiple histone proteins as done in panels G, H to validate their findings? Where do H2A, H3, H4 etc localize and is the overlap still perfectly correlative?

A strongly suggested experiment would be to perform IP-MS and IP-proximity MS to determine what are the other proteins that associate with MPO in stimulated/unstimulated neutrophils. This would complement the cryo-EM and further enable mutagenesis (discussed above/below), the authors should perform crosslinking type of MS to determine what other proteins may be directly interacting in the MPO-NCP network? This is useful and also would get the study closer to therapeutic relevance knowing what to target/what else is there, and better give the foundation for dispelling which mechanisms discussed in the discussion are most at play.

Figure 1G, H- please indicate protein MW on the left side of these figure panels.

Supplemental Figure 1B- why are the sizes of MPO different in the purified versus the cell lines? Is the upper band background? Also, why is there some MPO remaining in the delMPO cell line? With the size discrepancies, can the authors subject all bands to MS to confirm protein identity and sequence?

Regarding Figure 2/Supplemental- where is the figure showing the purified rMPO and the confirmation of reconstitution with nucleosomes? Where is the binding assay (I see just a schematic for Supp Fig S6). Would be helpful to see what went in to the cryoEM.

The structural work in Figure 2 is beautifully presented. As discussed above, extensive mutagenesis would be important here to evaluate various mutations, based on these structures, on MPO-nucleosome binding in vitro. Further, such mutations should be used to evaluate MPO binding in cells (i.e. all assays in Figure 1)? Impact of AP and auxiliary binding regions intact/mutated?

Relating to Figure 3 and 4, where is the purification of native MPO? Not detailed unless I missed it (S7a again only has a schematic—where are the gels and blots and confirmation of binding, etc?).

Can the authors make more clear why rMPO cannot dimerize/did not dimerize on NCPs but the MPO from human blood did? This isn't as clear as I think the reader would appreciate in the text. Can the authors also please show the differences in the sequences/splice variants, etc that could contribute to this?

Further to this, if the authors make a mutation (again mutagenesis is needed for this study) that breaks dimerization, what happens? I think this mutation is critical to evaluate the mechanism of dimerization and the impact on chromatin binding and nucleosome disassembly.

The authors use Widom 601 DNA for NCPs, which is standard and also very tightly bound. Can the authors examine the chromatin binding and disassembly using physiologic sequences found in NETs (i.e. isolate and sequence DNA from NETs)? This would make their findings more relevant.

While it is nice that the authors recovered different species—i.e. free nucleosomes and MPO monomer or nucleosome complexes—the absence alone of MPO dimer/nucleosome particles seems only suggestive but not proof of the hypothesis that MPO dimers are needed for nucleosome disassembly. Again here, dimer mutants are needed to confirm (i.e. keep concentrations the same but show no dimerization within human MPO.)

Finally, in Figure 5, I am not convinced by the data in Figure 5A/B, that this suggests colocalization of MPO with NETs. I don't think this adds much beyond what is in Figure 1. I suggest taking Figure 5C and adding it to Figure 1, since the theme and findings are the same—that MPO binds nucleosomes, whether from stimulated cells or CF sputum samples.

Other questions:

Are there any cell lines or model systems bearing mutations in MPO that the authors could use? Can the authors use recombinantly expressed mutant nucleosomes and overexpressed MPO to test interactions further?

Can the authors give more details re how they think their findings could be exploited therapeutically? This is touched upon in the discussion, but with the ambiguity re: which mechanism(s) are dominant here, this is hard to follow.

Version 1:

Reviewer comments:

Referee #1

(Remarks to the Author)

The revised study is improved, though some of the points that were raised still need to be addressed.

Major points

1. The first major point was about the requirement for MPO, which is presented in the Introduction (end of 3rd paragraph) as if it were a general feature of NETosis whereas it is clearly stimulus-dependent, based on existing studies from their own lab (such as PMID 28574339, which had specifically been mentioned). In their reply, the authors mainly focused on the interesting topic of how MPO-deficient patients would have to be better characterized in future publications, and do mention something about this in the revised Discussion. While this certainly has an incidence on the point being raised, it doesn't address it. As a result, it would still be more accurate to state that MPO is required *for some* NET stimuli but not others.

2. The second point was about whether MPO can associate with nucleosomes with a discontinuous distribution that co-localizes with nucleosomes, in response to biologically relevant stimuli. The authors did some run-of-the-mill MPO stainings using 3 such stimuli (TNF α /heme, MSU crystals, leukocidin) to show in Suppl Fig 5 that unsurprisingly, the resulting NETs

are MPO-positive (as all known NETs are). However, Suppl Fig 5 does not show the periodicity in MPO staining that is observed with the initial stimuli (PMA, nigericin) and indicative of preferential (or exclusive) nucleosomal binding. In their reply, the authors also mention that they detected both MPO and histones in isolated nucleosomes from MSU-derived NETs (Suppl Fig 6A). This does indeed answer a part of the point raised (at least for MSU, not for the other stimuli) insofar as it shows MPO association with nucleosomes. However, short of blotting the non-nucleosomal fraction of MNase digests for MPO, it does not reveal whether MPO might also be associated with NETs in between nucleosomes and thus display continuous binding.

As a result, Suppl Fig 5 should be removed because it only shows (in the authors own words) MPO-DNA colocalization, i.e. something that isn't new and that wasn't asked for. The most straightforward way of answering the point that was raised, would be to show the discontinuous staining pattern for MPO on NETs generated in response to biologically relevant stimuli, just as they did for PMA and nigericin. As an added benefit, such a demonstration would be nicely supported by the data in Suppl Fig 6A. Another way of answering this point (though it would be more labor-intensive) would be to blot both the nucleosomal and non nucleosomal fractions of MNase digests with Abs against MPO and histones, for each biological stimulus.

3. Major points 3 and 4 were satisfactorily answered.

4. All minor points were also satisfactorily answered.

Referee #2

(Remarks to the Author)

My main experimental concerns are addressed — this is an exciting study and a tour de force across disciplines. In particular, I found the experiments involving the chemical reduction of MPO into a monomer convincing.

I would still like to caution the authors and reiterate several points from my previous review: (i) Nucleosomes are inherently dynamic, and while the potential steric clashes between DNA and MPO dimers are noted, many nucleosome structures lack duplex density at the ENTRY/EXIT sites. Caution is therefore warranted in interpreting these observations; (ii) I remain unconvinced by the term 'DNA-transformer'—while creative, it feels like an over-interpretation of the data; (iii) Finally, using cryo-EM to quantitatively assess complex stability is problematic. Heterogeneous particles often fail to produce stable class averages and may thus be undercounted or missed entirely.

Referee #3

(Remarks to the Author)

The authors have worked extensively to address, as best they can, the points of this reviewer and the other two reviewers. In general, I appreciate their efforts but still find troubling the following three points (which hinder the mechanistic rigor and its impact that one can take away from this work):

First, the new studies (discussed on p13 of the response to reviewers) that aim to address the binding of MPO to the nucleosome acidic patch and of DNA—it sounds from the authors' response that they too cannot tell whether it is one or the other or both (multiple sites)—if multiple sites, again, I think it would be key in a paper of this level to understand the mode of interaction at that level of detail. I also do think mutagenesis studies ARE still imperative here, and can be done as single mutants and double-/multi-mutants. The antibody experiment does not quite get there...

Also, for MPO, they can make His-tagged recombinant MPO so they should be able to generate mutants on that side as well. Not sure why this cannot be done.

Second, page 16- I still think a simple stain, as is conventional in the field, for each histone would still be needed.

Finally, the authors did not pursue my suggestion of IP-MS experiments of NETs, which are absolutely able to be done and could/should have been done with WT and MPO mutants. This is important to make any statement about whether the MPO - NCP interaction facilitates specific (and interesting) interactions within the network.

Version 3:

Reviewer comments:

Referee #1

(Remarks to the Author)

The revised study duly addressed all remaining issues.

Referee #2

(Remarks to the Author)

I would stand by my previous comments:

"My main experimental concerns are addressed — this is an exciting study and a tour de force across disciplines. In particular, I found the experiments involving the chemical reduction of MPO into a monomer convincing.

We thank the reviewer for these positive comments.

I would still like to caution the authors and reiterate several points from my previous review:

(i) Nucleosomes are inherently dynamic, and while the potential steric clashes between DNA and MPO dimers are noted, many nucleosome structures lack duplex density at the ENTRY/EXIT sites. Caution is therefore warranted in interpreting these observations; (ii) I remain unconvinced by the term 'DNA-transformer'—while creative, it feels like an over-interpretation of the data; (iii) Finally, using cryo-EM to quantitatively assess complex stability is problematic. Heterogeneous particles often fail to produce stable class averages and may thus be undercounted or missed entirely."

Regarding the responses:

(i) I would encourage the authors to check the spontaneous dynamics of DNA unwrapping in PMID: 29323273. I understand the steric clash argument and think the additional comments added to the manuscript are helpful and have addressed my point. However, arguing about the ends of DNA in nucleosomes is a slippery slope.

(ii) Again, staying with the PMID reference, there seems to be some rather spontaneous "transformer" action happening on nucleosomes by thermal fluctuation. I understand the advantage of naming a process to make it referable, but I don't find this term helpful here. This is a binding process, no conceptually new transformation.

(iii) I disagree with the authors. Only shared conformations will ensemble average in cryo-EM in either 2D or 3D. It's impossible to quantify a structurally diverse intermediate in cryo-EM by averaging (quantitatively or qualitatively). Furthermore, some conformation - in the same sample - might be less stable at the air/water interface than others and hence not average; so the same proteins in cryo-EM will often not necessarily reflect the equilibrium state distribution. I'm aware that cryo-EM is used to argue for the equilibrium position in other publications, but I find it misleading. It doesn't change the mechanism much, agreed, but I don't follow the argument made. I would take the authors up on the suggestion to tone this down in the manuscript?

Referee #3

(Remarks to the Author)

The authors have satisfactorily addressed my concerns and those of the other reviewers. Thank you for the opportunity to review this exciting new work.

From: Arturo Zychlinsky <zychlinsky@mpiib-berlin.mpg.de>

Dear Ursula,

Hope you had a good trip.

Sorry to bother you again. I was wondering whether you had a chance to take a look at our point by point reply to the reviewers.

I am sending you here my previous email and attached the reply as well.

With best regards,

Arturo

On 11. Jun 2025, at 15:34, Arturo Zychlinsky <zychlinsky@mpiib-berlin.mpg.de> wrote:

Dear Ursula,

I hope that your trip is pleasant. Many thanks for the opportunity to reply to the reviewers.

Below is a summary of the referee's points (underlined) and our reply. I am also attaching our detailed point-by-point answers, including new data and a suggestion of the changes we can implement in the manuscript.

I would be very grateful for your feedback which of our suggested modifications should be included in a new version of the manuscript.

Best regards,

Arturo

Referee 1

-

1. Differences in the requirement for MPO activities.

In this manuscript we describe how MPO nucleosome disassembly depends on its binding to the acidic patch, but not on its enzymatic activity. In contrast, the enzymatic activity is important to start NETosis in response to some, but not all, stimuli. We can modify the introduction to clarify this point.

2. Discontinuous distribution of MPO in response to other NET stimuli

We now include data showing the periodic distribution on MPO in NETs induced by Panton Valentine Leukocidin (PVL) by SIM super-resolution. Together with the biochemical demonstration that MPO co-migrates with histones in MSU induced NETs (Supplementary Fig 6 and reply to reviewers).

-

Referee 2

-

(i) Take into account that nucleosomes are inherently dynamic

We agree with the reviewer that there are instances where DNA end displacement does not lead to complete nucleosome disassembly. However, in our system, we think that our structural and biochemical data collectively point strongly towards DNA end displacement being the driving force of nucleosome disassembly. But we suggest to modify the text to take the reviewer's concern into account.

(ii) Reconsider the term "transformer" for MPO.

We show that MPO converts chromatin, from its hereditary and transcriptional role, into an immune-effector. This is a novel function and we thus suggest to keep the term "transformer" to define the first of potentially several proteins with this function and distinguish it from other chromatin binding proteins.

(iii) Using cryo-EM is qualitative.

We agree with the reviewer that EM is a qualitative technique and can miss some rare "states" of the complex. We can rephrase the text to indicate this possibility.

-

Referee 3.

-

Generate MPO mutants.

To further describe the binding between MPO and nucleosomes the reviewer asked to generate MPO mutants. To recapitulate our structural findings; The MPO-nucleosome complex has two protein-protein interfaces between MPO monomer 1 and the nucleosome, and a protein-DNA interface between MPO monomer 2 and the nucleosome.

Because of the nature of the interaction, these experiments would not be conclusive for the following reasons:

- The protein-DNA interface cannot be mutated from the DNA side since MPO binds to the backbone phosphates, and is thus sequence-unspecific.
- Mutating the acidic patch or the auxiliary interface will likely prevent MPO from binding to the histones (completely or partially), but MPO will retain its ability to bind to the DNA (since it can also bind to DNA outside the context of nucleosomes). We tested this, as we show in our point-by-point response to the first round of reviews (page 14). There, we use nucleosomes with mutation in the acidic patch, and see only a mild effect on the binding kinetics as most likely MPO still binds to the DNA.
- Mutating MPO arginine anchors affect both binding to the acidic patch and to DNA. Hence, results with these mutants will be inconclusive.

-

To address the concern of the reviewer about MPO binding to the acidic patch which cannot be answered with mutations, we blocked MPO binding to the acidic patch with an antibody fragment. In this experiment, in contrast to the suggested mutations, we can effectively prevent MPO from positioning itself in the orientation that allows it to displace the DNA end. We observed that this antibody fragment blocks nucleosome disassembly (Supplementary Fig. 13C). Disappointingly, the reviewer did not like this experiment, without giving an explanation.

Use more anti-histone antibodies for the periodicity analysis.

-

We include new data showing that MPO colocalizes with an antigen generated by cleaving histone H3 during NETosis. This answers the question about the colocalization of MPO and nucleosomes in NETs. We believe that using antibodies to specific histones might be misleading since NET nucleosomes are heterogenous and epitopes might not be recognized by “standard” antibodies.

IP-MS experiments of NETs with MPO mutants

-

The reviewer suggested to include IP-MS experiments of NETs derived from MPO mutants. All the experiments on NETs were done with neutrophils isolated from healthy volunteers, since there is no cell line that can undergo NETosis. The very rare patients with MPO mutations do not make NETs. Hence the experiment suggested cannot be done.

The focus of the manuscript is how MPO modifies chromatin to make NETs. We think that including data on the composition of NET-monomer nucleosomes would be distracting. However, in the reply to reviewers attached, we include proteomic data of nucleosomes purified by fractionation. This proteome avoids the potential artefacts of cross-linking and antibody specificity. Indeed, in the absence of a biochemical and biological validation, we believe this “only” to be a list of many proteins. However, if you think it enriches the manuscript we can make a new figure and include these data.

<Rebuttal_2 ms 2025-03-07478A.pdf>

On 10. Jun 2025, at 05:19, Ursula Weiss <u.weiss@nature.com> wrote:

Dear Arturo,

Please send me a point by point rebuttal in advance. I am currently on a break in Central Asia and will be in touch when back.

Best wishes,

Ursula

Sent from my iPad

On 10 Jun 2025, at 02:27, Arturo Zychlinsky <zychlinsky@mpiib-berlin.mpg.de> wrote:

Dear Ursula,

Many thanks for sending the comments of the third reviewer as well. We believe that we can answer most questions of the three reviewers, but a few points need clarification, which would be best solved in a phone call. If you can give me a couple of dates when you are available, I would be happy to talk about it.

Thanks, Arturo

On 28. May 2025, at 18:22, Ursula Weiss <u.weiss@nature.com> wrote:

Dear Arturo,

I am sorry for the protracted review process. We have received two reports which I have pasted below. Comments from our third reviewer have been delayed because of an illness in the family. I expect to receive them by the end of the upcoming weekend.

I will be in touch in due course,

All best wishes,

Ursula

Referee #1:

The revised study is improved, though some of the points that were raised still need to be addressed.

Major points

1. The first major point was about the requirement for MPO, which is presented in the Introduction (end of 3rd paragraph) as if it were a general feature of NETosis whereas it is clearly stimulus-dependent, based on existing studies from their own lab (such as PMID 28574339, which had specifically been mentioned). In their reply, the authors mainly focused on the interesting topic of how MPO-deficient patients would have to be better characterized in future publications, and do mention

something about this in the revised Discussion. While this certainly has an incidence on the point being raised, it doesn't address it. As a result, it would still be more accurate to state that MPO is required *for some* NET stimuli but not others.

2. The second point was about whether MPO can associate with nucleosomes with a discontinuous distribution that co-localizes with nucleosomes, in response to biologically relevant stimuli. The authors did some run-of-the-mill MPO stainings using 3 such stimuli (TNF α /heme, MSU crystals, leukocidin) to show in Suppl Fig 5 that unsurprisingly, the resulting NETs are MPO-positive (as all known NETs are). However, Suppl Fig 5 does not show the periodicity in MPO staining that is observed with the initial stimuli (PMA, nigericin) and indicative of preferential (or exclusive) nucleosomal binding.

In their reply, the authors also mention that they detected both MPO and histones in isolated nucleosomes from MSU-derived NETs (Suppl Fig 6A). This does indeed answer a part of the point raised (at least for MSU, not for the other stimuli) insofar as it shows MPO association with nucleosomes. However, short of blotting the non-nucleosomal fraction of MNase digests for MPO, it does not reveal whether MPO might also be associated with NETs in between nucleosomes and thus display continuous binding.

As a result, Suppl Fig 5 should be removed because it only shows (in the authors own words) MPO-DNA colocalization, i.e. something that isn't new and that wasn't asked for. The most straightforward way of answering the point that was raised, would be to show the discontinuous staining pattern for MPO on NETs generated in response to biologically relevant stimuli, just as they did for PMA and nigericin. As an added benefit, such a demonstration would be nicely supported by the data in Suppl Fig 6A. Another way of answering this point (though it would be more labor-intensive) would be to blot both the nucleosomal and non nucleosomal fractions of MNase digests with Abs against MPO and histones, for each biological stimulus.

3. Major points 3 and 4 were satisfactorily answered.

4. All minor points were also satisfactorily answered.

Referee #2:

My main experimental concerns are addressed — this is an exciting study and a tour de force across disciplines. In particular, I found the experiments involving the chemical reduction of MPO into a monomer convincing.

I would still like to caution the authors and reiterate several points from my previous review: (i) Nucleosomes are inherently dynamic, and while the potential steric clashes between DNA and MPO dimers are noted, many nucleosome structures lack

duplex density at the ENTRY/EXIT sites. Caution is therefore warranted in interpreting these observations; (ii) I remain unconvinced by the term 'DNA-transformer'—while creative, it feels like an over-interpretation of the data; (iii) Finally, using cryo-EM to quantitatively assess complex stability is problematic. Heterogeneous particles often fail to produce stable class averages and may thus be undercounted or missed entirely.

Point-to-point response to the reviewers' comments

We gratefully thank the reviewers for their positive feedback and insightful comments, which aided us to further improve the manuscript. Below is a point-by-point response to all comments and a detailed description of all changes we have made to our manuscript after considering their suggestions. The changes are highlighted in yellow in the revised text.

Referee #1

The revised study is improved, though some of the points that were raised still need to be addressed.

We thank the reviewer for the positive feedback and reply to her/his review below.

Major points

1. The first major point was about the requirement for MPO, which is presented in the Introduction (end of 3rd paragraph) as if it were a general feature of NETosis whereas it is clearly stimulus-dependent, based on existing studies from their own lab (such as PMID 28574339, which had specifically been mentioned). In their reply, the authors mainly focused on the interesting topic of how MPO-deficient patients would have to be better characterized in future publications, and do mention something about this in the revised Discussion. While this certainly has an incidence on the point being raised, it doesn't address it. As a result, it would still be more accurate to state that MPO is required *for some* NET stimuli but not others.

We agree with the reviewer and should further clarify that MPO has two functions in NET formation. First, MPO *activity* is relevant in the response to some, but not all, NET stimuli. Second, as we show in this manuscript, MPO *binding* to the nucleosome is required for NET formation in response to, likely, all inducers.

In more detail:

Several labs, including ours, showed that MPO peroxidase activity is required for NET formation in response to microbes and mitogens (PMA), but not when neutrophils are stimulated with Nigercin or Panton Valentine Leukocidin (PVL) (PMID: 37943843, PMID:

28574339, PMID: 20974672). Why some stimuli require MPO catalytic activity and others do not is not understood.

In this manuscript we show a second, novel, function of MPO; the disassembly of nucleosomes which requires binding to the acidic patch and is independent of its enzymatic activity. MPO binds to nucleosomes when we incubate neutrophils with stimuli that require (e.g. PMA) or do not require (e.g. Nigericin) MPO peroxidase activity to make NETs. Our data are consistent with the observation that nuclei isolated from various cell types expand when incubated with MPO and that this activity cannot be blocked with peroxidase inhibitors or the withdrawal of halides from media (PMID: 20974816). In these experiments, nucleus expansion depends on a peroxidase-independent function of MPO for which we provide a novel mechanism in our manuscript.

In summary, MPO has two functions. First, as a peroxidase it generates products that are relevant in the initiation of NETosis in response to some, but not all, stimuli, as well as in other neutrophil functions like phagocytosis. Second, MPO binding to the acidic patch is relevant in transforming chromatin into NETs. This second function is relevant for all the NET inducing stimuli we tested so far, including Nigericin and PVL (see also our answer to point 2 raised by this reviewer) as demonstrated microscopically and biochemically.

In order to make this more clear in the manuscript, we have now replaced the following sentence in the penultimate paragraph of the introduction, on p. 4:

from:

Importantly, MPO is required for NET formation by way of chromatin decondensation in response to NET-inducing stimuli and MPO-deficient individuals cannot form NETs.

to:

The enzymatic activity of MPO is necessary to proceed with NETosis in response to some stimuli. Here we show that MPO, but not its enzymatic activity, is required for chromatin decondensation during NET formation in response to all the stimuli we tested. Importantly, MPO-deficient individuals cannot form NETs.

2. The second point was about whether MPO can associate with nucleosomes with a discontinuous distribution that co-localizes with nucleosomes, in response to biologically relevant stimuli. The authors did some run-of-the-mill MPO stainings using 3 such stimuli (TNF α /heme, MSU crystals, leukocidin) to show in Suppl Fig 5 that unsurprisingly, the resulting NETs are MPO-positive (as all known NETs are). However, Suppl Fig 5 does not

show the periodicity in MPO staining that is observed with the initial stimuli (PMA, nigericin) and indicative of preferential (or exclusive) nucleosomal binding.

In their reply, the authors also mention that they detected both MPO and histones in isolated nucleosomes from MSU-derived NETs (Suppl Fig 6A). This does indeed answer a part of the point raised (at least for MSU, not for the other stimuli) insofar as it shows MPO association with nucleosomes. However, short of blotting the non-nucleosomal fraction of MNase digests for MPO, it does not reveal whether MPO might also be associated with NETs in between nucleosomes and thus display continuous binding.

As a result, Suppl Fig 5 should be removed because it only shows (in the author's own words) MPO-DNA colocalization, i.e. something that isn't new and that wasn't asked for. The most straightforward way of answering the point that was raised, would be to show the discontinuous staining pattern for MPO on NETs generated in response to biologically relevant stimuli, just as they did for PMA and nigericin. As an added benefit, such a demonstration would be nicely supported by the data in Suppl Fig 6A. Another way of answering this point (though it would be more labor-intensive) would be to blot both the nucleosomal and non nucleosomal fractions of MNase digests with Abs against MPO and histones, for each biological stimulus.

As indicated by the reviewer, we believe that the data showing the co-fractionation of MPO and histones in NETs induced by MSU are of relevance to confirm the association of MPO with nucleosomes on NETs induced by MSU (Supplementary Figure 6). We also agree that the old Supplementary Figure 5 might be redundant with published data and we have removed it from the manuscript.

Furthermore, to address the reviewers questions on the periodicity of MPO in NETs induced by other stimuli, we performed SIM superresolution images on neutrophils stimulated by PVL. As shown in the new Supplementary Figure 5, the distribution of MPO and nucleosomes (labeled with the PL2.3 antibody) is similar with both stimuli.

3. Major points 3 and 4 were satisfactorily answered.
4. All minor points were also satisfactorily answered.

Referee #2 (Remarks to the Author):

My main experimental concerns are addressed — this is an exciting study and a tour de force across disciplines. In particular, I found the experiments involving the chemical reduction of MPO into a monomer convincing.

We thank the reviewer for these positive comments.

I would still like to caution the authors and reiterate several points from my previous review: (i) Nucleosomes are inherently dynamic, and while the potential steric clashes between DNA and MPO dimers are noted, many nucleosome structures lack duplex density at the ENTRY/EXIT sites. Caution is therefore warranted in interpreting these observations; (ii) I remain unconvinced by the term ‘DNA-transformer’—while creative, it feels like an over-interpretation of the data; (iii) Finally, using cryo-EM to quantitatively assess complex stability is problematic. Heterogeneous particles often fail to produce stable class averages and may thus be undercounted or missed entirely.

(i) We agree with the reviewer that nucleosomes are dynamic and that partial unfolding of the DNA from the histone core might not, in all cases, result in complete disassembly of the whole nucleosome. However, in all our structures, the one striking difference between samples that are stable (as we know from the biochemical experiments) and those that are not stable and will be disassembled (i.e. in the presence of MPO dimers), is the displaced and disordered DNA end. Taking this into account, we think that the most logical explanation is that the displacement of the DNA end leads to nucleosome disassembly. We can not, of course, completely rule out another, unobserved mechanism. We will thus add the following sentences in paragraph 2 of the Discussion (p. 14 in the manuscript) to reflect the remaining doubt which the reviewer points out:

It is important to point out that nucleosomes generally display some inherent structural dynamics and DNA end unfolding does not in all cases necessarily lead to complete eviction. Furthermore, we cannot categorically rule out other mechanisms that might also contribute to nucleosome destabilization, e.g. intermediate states too transient and rare to be captured by cryo-EM. Strikingly, however, in all our structures we observe disordered ends in biochemically unstable samples and ordered ends in biochemically stable samples. The most probable explanation for these observations is that DNA end destabilization ultimately leads to nucleosome disassembly.

(ii) We prefer the term "transformer" since MPO is the first protein known to change the function of chromatin from a structure that ensures heredity and transcription into an immune

defense effector. To distinguish MPO from other chromatin binding proteins, we believe this function definition is warranted. We think it is relevant to distinguish the function of MPO with other chromatin modifiers, and could not think of a better term.

(iii) We agree with the reviewer that samples in EM, in this or any other study using cryo-EM, might not completely reflect the conditions in solution since different macromolecular species might have different propensities to enter the holes of the EM grid or because fragile complexes might disassemble during the rather harsh grid preparation procedures. Furthermore, our description of the macromolecular species in our different EM samples (e.g. the different time points of the time course experiments) is not a quantitative one. Instead, we describe the presence or absence of macromolecular species in a comparative manner (samples within an experiment have been prepared in the same session under identical conditions). Consequently, our analysis is a qualitative and not a quantitative one. Having stated this, however, it is possible that the macromolecular species we retrieved from our EM analysis might not be a complete representation of what is in the samples and we might have missed for example very fragile and transient complexes or intermediate states that are too rare to occur in sufficient number for cryo-EM data processing. But, even if this was the case, it would not change the overall mechanism that we propose. We propose to add a note of this to the second paragraph of the discussion (see reply to point (i) by this reviewer) to take into account this limitation.

Referee #3 (Remarks to the Author):

The authors have worked extensively to address, as best they can, the points of this reviewer and the other two reviewers. In general, I appreciate their efforts but still find troubling the following three points (which hinder the mechanistic rigor and its impact that one can take away from this work):

We thank the reviewer for their criticism and hope that the explanations provided below will help to clarify some of the raised concerns.

First, the new studies (discussed on p13 of the response to reviewers) that aim to address the binding of MPO to the nucleosome acidic patch and of DNA—it sounds from the authors' response that they too cannot tell whether it is one or the other or both (multiple sites)—if multiple sites, again, I think it would be key in a paper of this level to understand the mode of interaction at that level of detail. I also do think mutagenesis studies ARE still imperative

here, and can be done as single mutants and double-/multi-mutants. The antibody experiment does not quite get there...

Also, for MPO, they can make His-tagged recombinant MPO so they should be able to generate mutants on that side as well. Not sure why this cannot be done.

We agree with the reviewer that untangling the contribution of the individual interfaces could be informative if it was possible. However, due to the complex binding mode, creating mutants is in some cases impossible or it would not be very informative, since it would not allow us to look at the contributions of individual interfaces, as detailed below.

The interaction between MPO and nucleosomes is very complex. In the MPO dimer, the first monomer contacts the histones using two interfaces, the acidic patch and the auxiliary interface (Fig. 2 and 3). In addition, there is the protein-DNA interface that is mediated by the second MPO monomer (Fig. 3E). Furthermore, it is evident from our intermediate state structure (Fig. 4C) and our analysis by Electrophoresis Mobility Shift Assay (EMSA) and negative stain EM (Supplementary Fig. 9D,E) that MPO can also bind only to the DNA both in the context of the nucleosome and also to “naked” DNA. Importantly, we cannot disrupt the protein-DNA interface by changing the DNA, since it is mediated by arginines and lysines of MPO binding to the phosphate of the DNA backbone in a sequence-independent manner.

Conversely, because of the ability of MPO to bind to DNA alone, there will always be residual binding even if we completely or partially disrupt the protein-protein interfaces by mutations in the acidic patch. This is evident from the Bio-Layer Interferometry (BLI) traces that we provide in the answer to the first round of reviews (p. 14 of the point-by-point answer to the first review), and which we have for quick access added again here as Figure A, where a mutation in the acidic patch affects the binding only slightly and where it is impossible to know whether the mutated acidic patch still contributes in part to the remaining binding or whether this is now all mediated by the DNA site. It is also important to note that while the interaction of MPO with the acidic patch is likely kinetically very stable, the MPO-DNA interface is probably more transient and dynamic, but since there are many equivalent “binding sites” along the DNA phosphate backbone, this can also result in overall rather efficient and “tight” binding.

Mutations of the MPO arginine anchors that are mostly responsible for the acidic patch binding would not only affect interaction with the acidic patch, but also with the DNA (since they are involved in interactions of MPO monomer 2 with the DNA backbone as well as in the interaction between both monomers and the DNA in the intermediate state). Hence, mutations of the arginine anchors are unlikely to be a good tool to decipher the contribution of individual interfaces.

So, taken together, while it would be possible to make mutations in the arginine anchors of recombinant MPO monomers (not MPO dimers since they need to be isolated from human

blood and thus mutagenesis is impossible) or in the acidic patch, it is highly unlikely that the data could be analyzed to determine the contribution of individual interfaces to the overall binding.

BLI measurements using wildtype nucleosome

Biosensors: Ni-NTA

Ligand: rMPO-10xHis

Analyte: wildtype nucleosome

BLI measurements using mutant nucleosome (H2A E61A)

Biosensors: Ni-NTA

Ligand: rMPO-10xHis

Analyte: Nucleosome with H2A E61A mutation (Epiccypher #16-1029)

Figure A. BLI traces of nucleosomes (wildtype or acidic patch mutation H2A E61A) binding to immobilized, 10xHis-tagged rMPO.

Importantly, because of this inability to make meaningful mutations, we set up the alternative assay using acidic patch blockage by the antibody fragment (Supplementary Figure 13C). This experiment, although it does not investigate the technical contributions of interfaces to overall binding affinity, it makes a biologically more relevant and interesting point: the presence of the antibody fragment completely prevents MPO from binding to the acidic patch due to steric hindrance, and consequently MPO is not positioned in the orientation required to displace the DNA end (Supplementary figure 13C of main manuscript). Interestingly, this protects the nucleosome from DNA unfolding and demonstrates that the acidic patch interface and the correct positioning of MPO on the nucleosome is functionally essential. We are therefore convinced that this experiment is more valuable than a mutagenesis experiment could be.

Second, page 16- I still think a simple stain, as is conventional in the field, for each histone would still be needed.

We understand the point raised by the reviewer, but staining individual histones will likely be confusing because histone tails are cleaved by proteases during NETosis, so most conventional histone antibodies that recognise the histone tail do not bind NETs. We took advantage of this cleavage by now providing another stain using the 3D9 antibody which recognises a neo-antigen that is generated on histone H3 tails on NETs (PMID: 36282064). We stained NETs with the 3D9 and MPO antibody and performed SIM microscopy, demonstrating that MPO colocalizes with 3D9 in a similar fashion to our other nucleosome recognising antibody PL2.3 (Supplementary Figure 2F and G). We also performed control experiments where we cross-correlated the 3D9 antibody with PL2.3 and show that these two antibodies are highly colocalized, thus confirming that both these antibodies recognise nucleosomes similarly (Supplementary Figure 2G).

Finally, the authors did not pursue my suggestion of IP-MS experiments of NETs, which are absolutely able to be done and could/should have been done with WT and MPO mutants. This is important to make any statement about whether the MPO -NCP interaction facilitates specific (and interesting) interactions within the network.

The reviewer suggested IP-MS experiments of NETs derived from MPO mutants. All the experiments on NETs were done with neutrophils isolated from healthy volunteers, since there is no cell line that can faithfully undergo NETosis. The very rare patients with MPO mutations do not make NETs (PMID: 20974672). Hence the experiment suggested cannot be done.

Alternatively, we provide in this reply, a proteome analysis for native mononucleosomes derived from PMA or Nigericin digested NETs that were separated into 24 fractions over 10-30% sucrose gradients in a similar fashion to Figure 1G in the manuscript.

We analyzed each fraction by mass spectrometry. Peptides were extracted using FragPipe and maxLFQ values were then used to assess peptide abundance within each fraction. We first analysed all peptides identified in each fraction using principal component analysis, and showed that fractions from sucrose separated mononucleosomes PMA or Nigericine cluster together, demonstrating for the first time that both these stimuli lead to a similar output with regard to proteins that decorate nucleosomes on NETs (figure B).

In addition, in an in depth analysis, employing k-means clustering, we distinguish different groups of proteins that cluster similarly across fractions. We identified 5 clusters. The data

were then assigned z-scores which allowed us to visualize the relative abundance distribution of a single protein across the fractions (figure C) These data provide the first roadmap of native nucleosomes isolated from NETs and separated by density, that in turn can be used as a roadmap for future studies in which other chromatin transformers will be identified.

Figure B: Principal component analysis of all peptides detected in fractions from PMA and Nigericin induced NETs.

We believe that this approach, which avoids the potential pitfalls of crosslinking and antibody specificity, is a more direct approach to describe the protein composition of NETs than the IP-MS suggested by the reviewer. However, in the absence of a biochemical and biological validation, we believe this “only” to be a list of many proteins. Furthermore, further characterization would go beyond the scope of this study and would delay the manuscript for many months, without contributing to the main point of our report. However, if the reviewers think it enriches the manuscript we can make a new figure and include these data.

Figure C: K-means clustering of both Nigericin and PMA identified 5 clusters. Each protein is expressed here as relative abundance across fractions derived from maxLFQ values.